# Structural insights into i-motif DNA structures in sequences from the insulin-linked polymorphic region

Dilek Guneri [1,5], Effrosyni Alexandrou[1,5], Kamel El Omari [2], Zuzana Dvořáková[3], Rupesh V. Chikhale[1], Daniel T. S. Pike[1], Christopher A. Waudby [1], Christopher J. Morris [1] ✉, Shozeb Haider [1,4] ✉, Gary N. Parkinson [1] ✉ & Zoë A. E. Waller [1] ✉

The insulin-linked polymorphic region is a variable number of tandem repeats region of DNA in the promoter of the insulin gene that regulates transcription of insulin. This region is known to form the alternative DNA structures, i-motifs and G-quadruplexes. Individuals have different sequence variants of tandem repeats and although previous work investigated the effects of some variants on G-quadruplex formation, there is not a clear picture of the relationship between the sequence diversity, the DNA structures formed, and the functional effects on insulin gene expression. Here we show that different sequence variants of the insulin linked polymorphic region form different DNA structures in vitro. Additionally, reporter genes *in cellulo* indicate that insulin expression may change depending on which DNA structures form. We report the crystal structure and dynamics of an intramolecular i-motif, which reveal sequences within the loop regions forming additional stabilising interactions that are critical to formation of stable i-motif structures. The outcomes of this work reveal the detail in formation of stable i-motif DNA structures, with potential for rational based drug design for compounds to target i-motif DNA.

Insulin (INS) is a protein hormone central to the regulation of glucose metabolism. Deficiencies or incorrect production of insulin can lead to hyperglycaemia and diabetes mellitus[1,2]. The insulin-linked polymorphic region (ILPR) is a variable number of tandem repeats (VNTR) region of DNA in the promoter of the insulin gene that regulates the transcription of insulin[3,4]. The ILPR is located 363 bp upstream of the insulin transcription start site, with heterogeneity in the number of tandemly repeated sequences observed among individuals[4]. The predominant ILPR sequence is composed of 14 base-pair tandem repeats comprising of 5'-ACAGGGGTGTGGGG-3'/3'-TGTCCCCACACCCC-5'[4]. Shortening in the length of the ILPR and variants in the sequences have been linked to the development of both Type-1 and Type-2 diabetes[5–9].

The ILPR influences both the expression of INS and insulin-like growth factor 2, and genetic variations in the ILPR are also associated with decreased expression of INS[5,10,11]. It has also been shown that insulin itself binds the G-rich regions in the ILPR[12]. However, exactly how the changes in the ILPR cause alterations in the expression of the INS gene remains unclear.

The ILPR is a GC-rich region of DNA[3–5,13,14]. The C-rich sequence can form i-motif structures, comprised of two parallel stranded hairpins zipped together by intercalated, hemi-protonated cytosine-cytosine base-pairs[15]. In contrast, the G-rich sequence can fold into G-quadruplexes, formed from planar G-quartets, held together by Hoogsteen hydrogen-bonding and stabilised via π–π stacking and coordination of

[1]School of Pharmacy, University College London, 29-39 Brunswick Square, London WC1N 1AX, UK. [2]Diamond Light Source, Harwell Science and Innovation Campus, Chilton, Didcot OX11 0DE, UK. [3]Institute of Biophysics of the Czech Academy of Sciences, Královopolská 135, 612 00, Brno, Czech Republic. [4]UCL Centre for Advanced Research Computing, University College London, Gower Street, London WC1E 6BT, UK. [5]These authors contributed equally: Dilek Guneri, Effrosyni Alexandrou. ✉e-mail: chris.morris@ucl.ac.uk; shozeb.haider@ucl.ac.uk; gary.parkinson@ucl.ac.uk; z.waller@ucl.ac.uk

cations[16]. These types of non-canonical structures have been shown to exist in cells and are prevalent within the promoter regions of genes, in particular regulatory elements[17–19]. Although G-quadruplex structures have been well studied, much less research has focussed on i-motifs, the sequences that comprise them, the corresponding structures they form and their biological functions.

Herein we expand the characterisation of the predominant C-rich[20,21] and G-rich[22] ILPR sequence to 11 native ILPR variants[13]. A relationship between in vitro formation of i-motif and G-quadruplex structures within the ILPR and corresponding *in cellulo* reporter gene expression is determined. A crystal structure of an intramolecular i-motif also reveals that the sequences within the loop regions, and their additional stabilising interactions, are critical to formation of the stable i-motif structures.

## Results and discussion

### Characterisation of the sequence variants within the ILPR

The polymorphism in the length of the regulatory promoter region of the insulin gene was suggested as a genetic marker for non-insulin-dependent diabetes in 1983[23]. A follow-up population study with 298 unrelated individuals revealed that the 5′-flanking region of the human insulin gene is polymorphic in both nucleotide length and sequence[13]. Rotwein et al.[13] reported 14 sequence variants, with minor changes from the predominant ILPR sequence, but three variants were limited to only 0.2% of the population. An early study showed some ILPR variants were inserted as an isolated segment into a minimal prolactin promoter-luciferase construct and co-transfected with a known insulin-related transcription factor, Pur-1, to mimic beta-cell molecular microenvironment in non-beta cells. This showed that the over-expression of Pur-1 has different effects on gene expression in the minimal promoter system with different ILPR variants. The highest Pur-1 affinity was associated with the most prevalent ILPR sequence and provides the initial proof-of-concept that the ILPR is linked to regulating insulin expression[5]. Another study focused on three of the G-rich ILPR variants and was able to correlate a relationship between the conformation of the G-quadruplex structure with binding affinity to insulin and insulin-like growth factor[22]. Although the G-rich variants from the ILPR have been investigated, studies on the C-rich sequences are limited to the most prevalent variant[20,21,24,25].

Here we focused on the 11 main ILPR variants of both the C-rich (ILPRC, Table 1) and the G-rich sequences (ILPRG, Table 2) to fully understand the relationship between variant sequence, structure, and function. Each tandem repeat has two tracts of guanines/cytosines (5′-ACAGGGGTGTGGGG–3′/3′-TGTCCCCACACCCC–5′) so we designed our sequences to have two repeats, providing the minimum sequence necessary for i-motif or G-quadruplex formation. We maintained flanking sequences on either side of the terminal C/G-tracts in line with the tandem repeat sequence. The general sequence for each variant was Flank-(C/G-tract)-Loop$_1$-(C/G-tract)-Loop$_2$-(C/G-tract)-Loop$_3$-(C/G-tract)-Flank. Each C-rich and G-rich variant was characterised using circular dichroism (CD) to determine the overall topology, thermal difference spectroscopy (TDS) to characterise the type of structure in solution, and UV melting/annealing experiments to determine the thermal stability. For the C-rich sequences, the transitional pH was determined by CD, to allow comparison of the pH stability of the sequence variants.

The C-rich ILPR sequence variants were characterised in 10 mM sodium cacodylate buffer with 100 mM KCl. CD spectroscopy was performed at a range of pHs between 4 and 8 for determination of the pH$_T$. TDS and UV melting and annealing experiments were performed at pH 5.5 to allow for assessment of the relative stability of all sequences, even those that may not be stable at physiologically relevant pH, allowing all sequences to be compared alongside each other. A summary of the sequence, the melting temperature ($T_m$), annealing temperature ($T_a$), transitional pH (pH$_T$), and structural assignment by

TDS are provided in Table 1. The corresponding representative data is provided in the supplementary information (Supplementary Figs. 1–5).

The predominant ILPR C-rich variant (1C) gave clear UV melting and annealing traces (Supplementary Fig. 1) and a $T_m$ of $55 \pm 0.2\,°C$ (Table 1). This melting temperature is higher than that measured by others for the same sequence. Although the previous work was performed in phosphate buffer, which displays reduced buffering capacity at elevated temperatures[20], the likely cause of the different $T_m$ values is differences in annealing procedures[26]. The TDS of sequence 1C showed positive peaks at 240 and 265 nm and a negative peak at 295 nm, consistent with an i-motif structure signature profile (Fig. 1A)[27]. Similarly, CD spectroscopy of variant 1C showed i-motif formation at acidic pH, indicated by a positive peak at 288 nm and a negative peak at 260 nm (Fig. 1B and Supplementary Fig. 2)[28]. As the pH increases towards pH 7, the structure unfolds, the positive peak shifts to 273 nm and the negative peak to 250 nm (Supplementary Fig. 2A). The pH$_T$ of 1C was determined to be 6.6 (Table 1 and Supplementary Fig. 2F), in-line with previous experiments for this sequence[24]. Variant 4C demonstrated the same stability as 1C, with a pH$_T$ of 6.7 (Table 1, Supplementary Fig. 2I) but the other variants were all significantly less pH stable, with pH$_T$s as low as 4.6 and 5.1 (11C and 10C, Table 1 and Supplementary Fig. 4). Interestingly, minor differences in the sequence made significant changes in the stability of the structures formed. For example, in 2C, a single C-to-G mutation in each of the tandem repeats results in significantly lower melting ($50 \pm 0.6\,°C$ compared to $55 \pm 0.2\,°C$) and annealing temperatures ($p < 0.0001$) and also a lower pH$_T$ of 5.2 ($p < 0.0001$). It appears to be a hairpin-like structure at pH 5.5 in the TDS analysis (Fig. 1A) and from the CD spectra at pH 5.5 (Fig. 1B). I.e., this C to G mutation prevents i-motif formation.

Two main factors appear to decrease the stability of i-motif structure in these variants: mutation of loop nucleotides from cytosine to guanine (6C-11C, Table 1) or mutation/truncation within the C-tracts (2C, 3C, 5C, 7C and 10C, Table 1). Some sequences are affected by both these factors (2C, 7C and 10C, Table 1) and have some of the lowest pH$_T$s overall (5.2, 5.4 and 5.1, respectively). The C to G mutation is critical as it both removes cytosines from the core stack of base pairs and introduces potential competing Watson/Crick complementary nucleotides, which can shift the conformational equilibrium towards hairpin formation. This is further supported by the data acquired for variants 10C and 11C which have more guanines in the loops; both sequences did not give any transitions in the UV melting/annealing experiments at 295 nm (Supplementary Fig. 1D and Supplementary Fig. 1L) but did at 260 nm (Supplementary Fig. 1H, 1P), indicative of hairpin/duplex formation. TDS also indicated a spectrum inconsistent with i-motif and more consistent with duplex (Fig. 1A and Supplementary Fig. 5), suggesting that these sequences form only hairpins under these experimental conditions. In silico structural calculations of these sequences using M-fold[29], also show clear potential for these sequences to fold into hairpins (Supplementary Fig. 6). Given the vast differences in structures and stability of the C-rich sequences, we examined the complementary G-rich sequences, to determine any complementarity in structure formation.

The G-rich ILPR sequence variants were characterised by CD in 10 mM sodium cacodylate buffer, pH 7.0, with 100 mM of KCl, NaCl or LiCl, to reveal cation preferences typically observed in G-quadruplex forming sequences. Sequences were characterised by UV melting/annealing in analogous buffer except with 20 mM KCl (100 mM concentrations of KCl resulted in $T_m$ values > 95 °C). A summary of the characterisation of the sequences in KCl cation conditions: the melting temperature ($T_m$), annealing temperature ($T_a$), QGRS mapper score[30], and structural assignment by CD and TDS are provided in Table 2. Corresponding example data is provided in the supplementary information (Supplementary Figs. 7–9).

The predominant ILPR G-rich variant (1G) gave clear UV melting and annealing traces (Supplementary Fig. 7A) and a $T_m$ of $76 \pm 0.6\,°C$

**Table 1 | Biophysical characterisation data for the sequences of the C-rich ILPR variants**

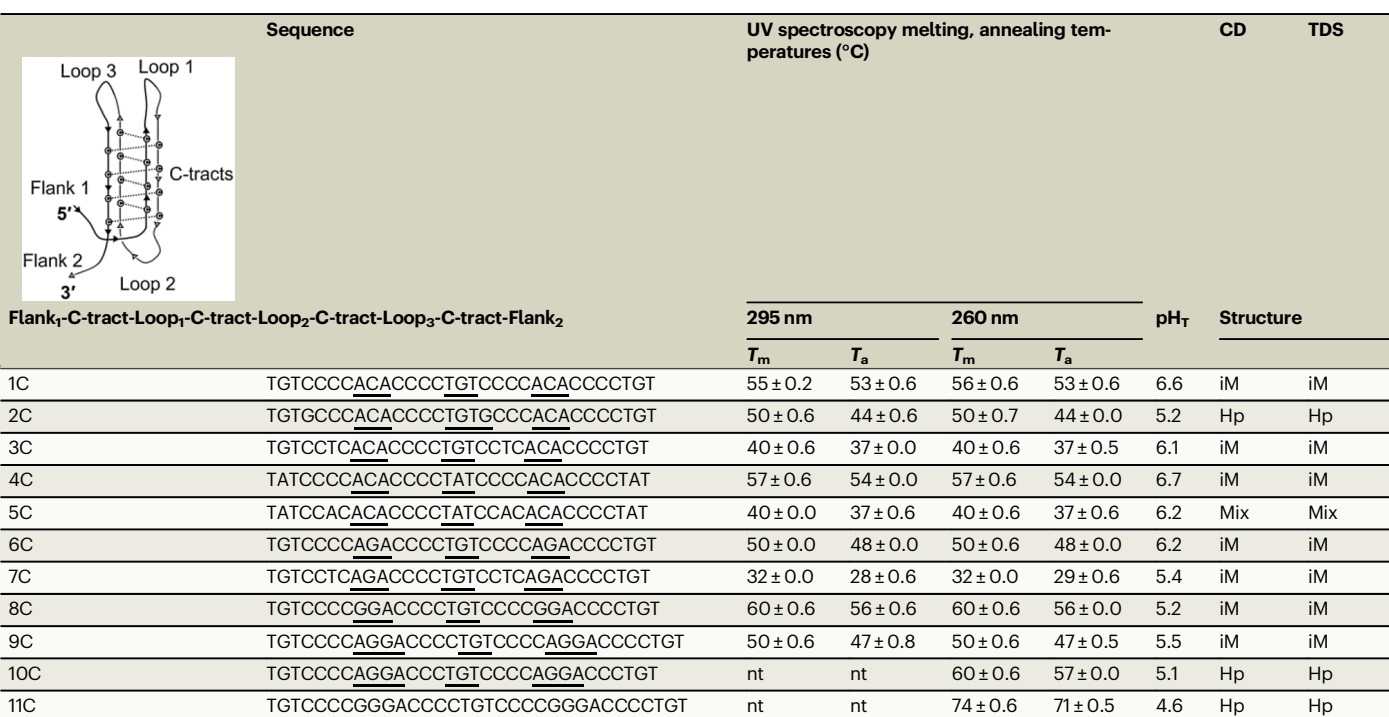

| Flank₁-C-tract-Loop₁-C-tract-Loop₂-C-tract-Loop₃-C-tract-Flank₂ | Sequence | UV spectroscopy melting, annealing temperatures (°C) | | | | | CD | TDS |
|---|---|---|---|---|---|---|---|---|
| | | 295 nm | | 260 nm | | $pH_T$ | Structure | |
| | | $T_m$ | $T_a$ | $T_m$ | $T_a$ | | | |
| 1C | TGTCCCCACACCCCTGTCCCCACACCCCTGT | 55 ± 0.2 | 53 ± 0.6 | 56 ± 0.6 | 53 ± 0.6 | 6.6 | iM | iM |
| 2C | TGTGCCCACACCCCTGTGCCCACACCCCTGT | 50 ± 0.6 | 44 ± 0.6 | 50 ± 0.7 | 44 ± 0.0 | 5.2 | Hp | Hp |
| 3C | TGTCCTCACACCCCTGTCCTCACACCCCTGT | 40 ± 0.6 | 37 ± 0.0 | 40 ± 0.6 | 37 ± 0.5 | 6.1 | iM | iM |
| 4C | TATCCCCACACCCCTATCCCCACACCCCTAT | 57 ± 0.6 | 54 ± 0.0 | 57 ± 0.6 | 54 ± 0.0 | 6.7 | iM | iM |
| 5C | TATCCACACACCCCTATCCACACACCCCTAT | 40 ± 0.0 | 37 ± 0.6 | 40 ± 0.6 | 37 ± 0.6 | 6.2 | Mix | Mix |
| 6C | TGTCCCCAGACCCCTGTCCCCAGACCCCTGT | 50 ± 0.0 | 48 ± 0.0 | 50 ± 0.6 | 48 ± 0.0 | 6.2 | iM | iM |
| 7C | TGTCCTCAGACCCCTGTCCTCAGACCCCTGT | 32 ± 0.0 | 28 ± 0.6 | 32 ± 0.0 | 29 ± 0.6 | 5.4 | iM | iM |
| 8C | TGTCCCCGGACCCCTGTCCCCGGACCCCTGT | 60 ± 0.6 | 56 ± 0.6 | 60 ± 0.6 | 56 ± 0.0 | 5.2 | iM | iM |
| 9C | TGTCCCCAGGACCCCTGTCCCCAGGACCCCTGT | 50 ± 0.6 | 47 ± 0.8 | 50 ± 0.6 | 47 ± 0.5 | 5.5 | iM | iM |
| 10C | TGTCCCCAGGACCCTGTCCCCAGGACCCTGT | nt | nt | 60 ± 0.6 | 57 ± 0.0 | 5.1 | Hp | Hp |
| 11C | TGTCCCCGGGACCCCTGTCCCCGGGACCCCTGT | nt | nt | 74 ± 0.6 | 71 ± 0.5 | 4.6 | Hp | Hp |

ILPRC sequences and their labels (1C-11C). Putative loops are underlined. Most prominent UV melting and annealing temperatures, transitional pH, and structure characterisation from the CD and thermal difference spectra (TDS) at pH 5.5. UV and CD spectroscopy were repeated in triplicate and data are shown as mean±0.02 (*n* = 3). *iM* i-motif, *Mix* Mixed species and hairpin, *Hp* hairpin, *nt* no transition observed.

**Table 2 | Biophysical characterisation data for the sequences of the G-rich ILPR variants**

| Sequence Flank₁-G-tract-Loop₁-G-tract-Loop₂-G-tract-Loop₃-G-tract-Flank₂ | UV spectroscopy melting, annealing temperatures (°C) | | | | CD | TDS | QGRS score |
|---|---|---|---|---|---|---|---|
| | 295 nm | | 260 nm | | | | |
| | $T_m$ | $T_a$ | $T_m$ | $T_a$ | Structure | | |
| 1G ACAGGGGTGTGGGGACAGGGGTGTGGGGACA | 76 ± 0.6 | 60 ± 0.0 | 77 ± 0.6 | 60 ± 0.6 | G4 | G4 | 63 |
| 2G ACAGGGGTGTGGGCACAGGGGTGTGGGCACA | 56 ± 0.0 | 50 ± 0.0 | 56 ± 0.6 | 51 ± 0.6 | Mix | Mix | 42 |
| 3G ACAGGGGTGTGAGGACAGGGGTGTGAGGACA | 30 ± 0.0 | 27 ± 0.6 | 31 ± 0.6 | 28 ± 0.6 | G4 | Hp | 21 |
| 4G ATAGGGGTGTGGGGATAGGGGTGTGGGGATA | 79 ± 0.6 | 60 ± 0.6 | 79 ± 0.6 | 60 ± 0.6 | G4 | G4 | 63 |
| 5G ATAGGGGTGTGTGGATAGGGGTGTGTGGATA | nt | nt | 52 ± 0.0 | 37 ± 0.6 | Hp | Hp | 21 |
| 6G ACAGGGGTCTGGGGACAGGGGTCTGGGGACA | 82 ± 0.7 | 56 ± 0.6 | 81 ± 0.6 | 57 ± 0.7 | G4 | G4 | 63 |
| 7G ACAGGGGTCTGAGGACAGGGGTCTGAGGACA | nt | nt | 51 ± 0.0 | 48 ± 0.6 | Hp | Hp | 21 |
| 8G ACAGGGGTCCGGGGACAGGGGTCCGGGGACA | 80 ± 0.7 | 60 ± 0.6 | 81 ± 0.6 | 60 ± 0.0 | Mix | Mix | 63 |
| 9G ACAGGGGTCCTGGGGACAGGGGTCCTGGGGACA | nt | nt | 57 ± 0.6 | 51 ± 0.6 | Hp | Hp | 62 |
| 10G ACAGGGTCCTGGGGACAGGGTCCTGGGGACA | nt | nt | 54 ± 0.6 | 51 ± 0.0 | RC | Hp | 42 |
| 11G ACAGGGGTCCCGGGGACAGGGGTCCCGGGGACA | nt | nt | 68 ± 0.0 | 63 ± 0.0 | Hp | Hp | 62 |

ILPRG sequences and their labels (1G-11G). Putative loops are underlined. All biophysical analyses were performed at pH 7.0. Most prominent UV melting, annealing and hysteresis temperatures, and structure characterisation from the CD and thermal difference spectra (TDS), and QGRS Score[30]. UV and CD spectroscopy were repeated in triplicate and data are shown as mean ± SD (*n* = 3). G4 = G-quadruplex; *Mix* Mixed species, *Hp* hairpin, *nt* no transition observed.

(Table 2). This melting temperature is similar to that measured previously (~78 °C) for the same sequence in Tris buffer[31]. The TDS of sequence 1G showed several positive peaks at 240, 255 and 270 nm and a negative peak at 295 nm, consistent with G-quadruplex structure[27] (Fig. 1C). Variant 1G gave CD spectra with a negative peak at 245 nm, and positive peaks at 263 nm and 295 nm (Fig. 1D). This is in-line with previous CD spectroscopy data, showing a mixed population of parallel and antiparallel G-quadruplexes in the presence of KCl and a shift towards antiparallel G4 formation in the presence of weaker stabilising cations NaCl and LiCl (Supplementary Fig. 8A)[14,31,32].

Of the other G-rich sequence variants, 4G ($T_m$ = 79 ± 0.6 °C) had similar thermal stability compared to 1G (76 ± 0.6 °C), showing the mutation in G-tract-Loop₂ from C to T makes little difference in the stability (Supplementary Fig. 7B, F). Variant 8G was more stable (80 ± 0.7 °C) than 1G, but does present as a potential mixture of species by CD and TDS (Supplementary Figs. 7K, 8G, and 9). The most stable of all the variants was sequence 6G (82 ± 0.7 °C). 1G, 4G, and 6G were all clearly characterised as G-quadruplexes by CD and TDS, similar to previously studies on these sequences (Supplementary Figs. 8 and 9)[22,32]. Other sequences formed significantly weaker DNA

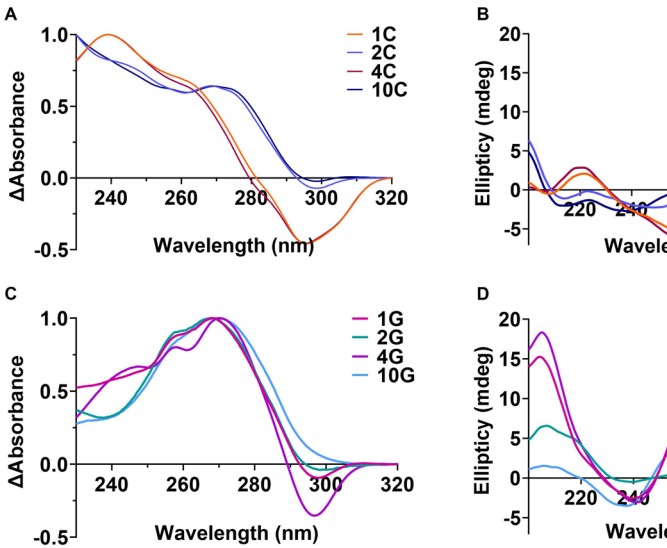

**Fig. 1 | Representative circular dichroism and thermal difference spectroscopy of ILPR variants examined in the cell-based reporter gene assay. A** TDS of 2.5 μM ILPRC variants 1C, 2C, 4C and 10C in 10 mM sodium cacodylate, 100 mM KCl at pH 5.5. **B** CD spectra of 10 μM ILPRC variants 1C, 2C, 4C and 10C in 10 mM sodium cacodylate, 100 mM KCl at pH 5.5. **C** TDS of 2.5 μM ILPRG variants 1G, 2G, 4G and 10G in 10 mM sodium cacodylate, 20 mM KCl at pH 7.0. **D** CD spectra of 10 μM ILPRG variants 1G, 2G, 4G and 10G in 10 mM sodium cacodylate, 100 mM KCl at pH 7.0. Source data for this figure are provided as a Source Data file.

structures and were thermally less stable than these strong G-quadruplexes. For example, 2G has a significantly lower melting ($T_m = 56 \pm 0$ °C compared to the 1G variant with a $T_m$ of $76 \pm 0.6$ °C) and annealing temperatures ($p < 0.0001$) and presents as a mixture of G-quadruplex and hairpin-like structures in the TDS (Fig. 1C). CD spectra show a broad weak positive peak at 300 nm and a negative peak at 245 nm (Fig. 1D), in-line with formation of a weak antiparallel G-quadruplex, potentially mixed with hairpin/duplex[28]. The fact that there is a melting transition in the UV at 295 nm (Supplementary Fig. 7I) potentially indicates G-quadruplex DNA structure, however, a negative peak at 295 nm in the TDS may present with Z-DNA and Hoogsteen DNA as well as G-quadruplex and i-motif structures[28]. Interestingly, some of the sequences (5G, 7G, 9G, 10G, and 11G) do not have UV melt and anneal profiles at 295 nm, as expected with G-quadruplex structures. However, variants 7G, 9G, 10G, and 11G have clear melting and annealing transitions at 260 nm, consistent with these sequences forming hairpins/duplex-like structures (Supplementary Fig. 7)[33]. For example, the 10G variant, is similar to 2G in the TDS signature (Fig. 1C), consistent with hairpin formation and clearly different to that of the G-quadruplexes formed by 1G, 4G, and 6G (Supplementary Fig. 8). The CD spectrum of 10G shows only a very weak positive signal at 260 nm and a negative signal at 240 nm (Fig. 1D), which is consistent with unfolded G-rich sequence or a very weak G-quadruplex. Notably, the hairpin-forming variants lose cation sensitivity in the CD spectra and all have a narrow dip in signal at 215 nm (Supplementary Fig. 8). M-fold predictions show clear potential for these sequences to form into hairpins (Supplementary Fig. 10).

These results indicate that only some ILPR variants are capable of forming i-motif and G-quadruplex structures. Comparing the biophysical data with the G-scores from QGRS Mapper[30] (Table 2) indicates that QGRS Mapper accurately predicts the most stable G-quadruplex forming sequences 1G, 4G, 6G and 8G (all score 63), but there are two sequences that score nearly as high (9G and 11G, score 62) that do not form G-quadruplexes at all. Moreover, it is important to consider that sequences such as 2G and 10G do not form stable G-quadruplex structures, but score 42, the same score as the G-quadruplex forming sequence from the widely-studied human telomere (TTAGGGT TAGGGTTAGGGTTAGGGTTA), demonstrating that loop nucleotide composition is critical.

The biophysical data for the C-rich and G-rich ILPR variants show that stable i-motifs are not exclusively formed in the complementary sequences of stable G-quadruplexes. From the native variants, 7/11 of the C-rich sequences form stable i-motif structures, whereas only 3/11 of the G-rich sequences form clear G-quadruplex structures. This indicates that it may be easier to mutate out a G-quadruplex based on the sequence, whereas an i-motif structure is more difficult to eliminate completely.

## DNA structures switch insulin reporter gene transcription

From the biophysical data, it was clear that not all native ILPR variants form i-motif or G-quadruplex structures. Importantly, the most common variants (1C and 4C) formed the most stable i-motif structures, and the complementary strands (1G and 4G) also formed stable G-quadruplex structures. We hypothesised that the DNA sequences forming into i-motifs and G-quadruplexes in the ILPR are potentially binding elements to control transcription of insulin. To test this hypothesis, we compared four ILPR variants using a Luciferase-based reporter gene assay, where the entire human insulin promoter up to the start of the ILPR was cloned upstream of the gene encoding firefly luciferase[34]. Resulting firefly bioluminescence is proportional to the insulin promoter activation. Due to the difficulty in cloning long lengths of the ILPR and the fact that this region of DNA is intrinsically variable between people, we included enough repeat sequences to form one i-motif or G-quadruplex. We chose the most common variants (1C/1G and 4C/4G), which formed both stable i-motif and G-quadruplex structures and two of the variants that appeared to form hairpin structures in both C-rich and G-rich sequences (2C/2G and 10C/10G).

Functioning β-cells normally secrete insulin in response to increased blood glucose levels as part of blood glucose homoeostasis. There are many cell line models which can be used to assess levels of insulin expression in vitro. These cells retain normal regulation of glucose-induced insulin secretion, allowing the use of glucose as a positive control[35]. We selected the rat insulinoma-derived cell line INS-1 as model system due to the lack of an intrinsic ILPR or analogous sequence[36,37]. The INS-1 cells were co-transfected with either one of the ILPR vectors and a reference vector encoding *renilla* luciferase to allow normalisation of transfection efficiency variability between experiments. After transfection, the cells were starved overnight and

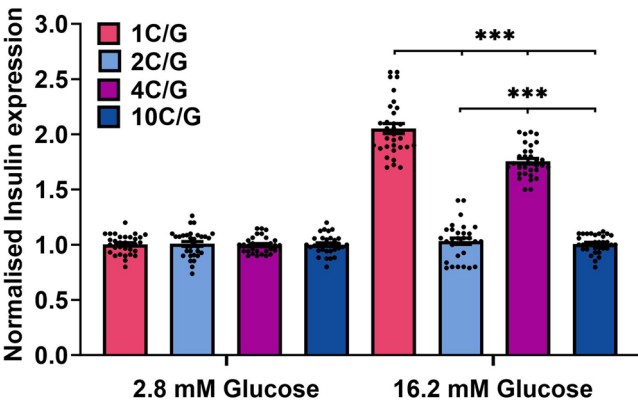

**Fig. 2 | Dual Luciferase-reporter gene assay for glucose sensitivity in co-transfected INS-1 cells after four hours.** Firefly signal is regulated by the human insulin promotor and is corrected to reference renilla luciferase signal. Firefly to renilla ratio is normalised to luminescence signals in low glucose levels, to represent insulin expression induced by glucose. Relative insulin reporter expression was determined in four different ILPR variants (1C/G, 2C/G, 4C/G or 10C/G), measured in 12 biological repeats ($n = 12$), each with 2–3 technical repeats and expressed in Mean ± SEM. All samples passed the D'Agostino & Pearson test for normal distribution. Statistical analysis was performed using 2-way ANOVA multiple comparisons with Holm-Šidák post hoc test. $p < 0.001$***, ns > 0.12. Source data for this figure are provided as a Source Data file.

were treated with either fresh low (2.8 mM) or high glucose (16.2 mM) medium to determine their respective responsiveness to glucose after four hours. These high/low glucose treatment conditions are consistent with other previous studies measuring responsiveness to glucose[34,38,39].

The four ILPR variants showed no significant difference in firefly luciferase expression in the presence of low (2.8 mM) glucose levels (Fig. 2). However, in the presence of high glucose (16.2 mM), there was a significant increase in the expression of luciferase relative to the control for the 1C/G and 4C/G plasmid variants (where the underlying sequences were shown to form stable i-motif and G-quadruplex structures) and no significant change in expression for the 2C/G and 10C/G plasmid variants (characterised to form hairpin-like structures). Specifically, the plasmid containing the 1C/G ILPR variant sequence showed a twofold increase in firefly luciferase expression levels ($p < 0.001$) in the presence of high glucose compared to low glucose concentrations. It was expected for the most prevalent ILPR sequence (1C/G) to show glucose responsiveness and therefore we considered as the positive control for this system. The plasmid containing the 4C/G ILPR variant sequence also responded to the higher glucose level with a 1.7-fold increase in gene expression compared to low glucose levels ($p < 0.001$). Both example ILPR variants, with sequences capable of forming i-motifs and G-quadruplexes responded to changes in glucose levels in a similar fashion, but the increase was significantly higher in the most prevalent ILPR variant (1C/G, $p < 0.001$). Importantly, reporters encoding the two ILPR variants that did not form i-motif and G-quadruplex DNA structures (2C/G and 10C/G) showed no changes in the presence of high glucose (Fig. 2). These data indicate the potential importance of the different sequence variants in the ILPR, showing the different DNA structures they form may play a role in controlling the responsiveness to glucose. Although the plasmid experiments do not directly assess transcription in endogenous chromatin, they do imply that only small changes in sequence can give rise to a very big difference in the structure formed and also the relative reporter expression in plasmids.

### Determination of an intramolecular i-motif crystal structure
Given that small differences between DNA sequences resulted in different structure formation in vitro and potential function in the

reporter genes *in cellulo*, we were interested in the potential interactions within the loops that made certain sequence variants more stable than others. The most biologically relevant DNA structures are intramolecular, i.e., those formed from a single strand, similar to what would form in the context of genomic DNA. However, structural information on intramolecular i-motifs is particularly scant. Although there are intramolecular NMR structures for i-motif (1EL2, 1ELN)[40], they are of modified fragments from the telomeres, and these modifications (necessary to enable structure determination by NMR) have been shown to alter the widths of the grooves in the structure[41]. There are currently twelve intermolecular i-motif crystal structures formed from two or four separate strands, but no intramolecular topologies. The apparent reason for the lack of intramolecular crystal structures is mainly due to the fact that i-motif loops are highly dynamic and difficult to resolve successfully using crystallographic methods. Intramolecular i-motif crystal structures would provide an opportunity for rational design of compounds to target these structures, and potential for drug development against these interesting biological targets, complementing drug discovery projects targeting G-quadruplex.

With this in mind, we wanted to give the best chance for successful crystallisation, so we trialled the most stable C-rich ILPR variant that formed only i-motif from our biophysical studies: (4C) TATCCCC ACACCCCTATCCCCACACCCCTAT. This sequence is the second most prevalent ILPRC variant[13] and lacks guanines within the sequence, so it reduces the formation of intermolecular species through GC-base-pairing. To increase the chance of successful crystallisation we also designed variants of this sequence with different flanking regions: TCCCCACACCCCTATCCCCACACCCCT (4Ca) and ATCCCCACACCC CTATCCCCACACCCC (4Cb) (Supplementary Table 1). The crystallisations were performed at pH 5.5, below the pHT, where this sequence would be most stable.

Crystals of all three variants were obtained by hanging-drop methods (Supplementary Table 2) with the highest-quality diffraction data acquired with 4C. With the limited availability of i-motif structures, molecular replacement (MR) methods proved challenging. However, anomalous dispersion (AD) methods were successful in structure determination and model validation using both intrinsic and extrinsic scattering elements. Intrinsic phosphorous single–wavelength anomalous dispersion (P-SAD) where phosphorus is integral part of the DNA backbone provided validation of the native structural model (Supplementary Fig. 11) while the use of extrinsic bromine, combined with multiple-wavelength anomalous dispersion (Br-MAD) methods provided anomalous scattering sufficient to generate high-quality maps for model building (Supplementary Tables 3 and 4). In the 4C-Br sequence (Supplementary Table 1), the Br-substitution located within the less flexible CC-core (Cytosine-4) provided a strong anomalous scattering contribution, while scattering for the second bromide loop-2 (Adenine-16) was not observed due to the flexibility of this region.

The general use of intrinsic P-anomalous scattering for structure determination of DNA/RNA motifs has proven challenging. The post analysis of our long-wavelength data revealed a limited P-anomalous scattering contribution, resulting in only a few P-peaks of the anomalous difference map ($F_{anom(calc)}$) overlapping the modelled positions or those observed with only weak diffuse peaks, this is despite using the lower energies closest to the peak (3.9995 Å, f"2.3). The poor P-signal can be partly attributed to the static disorder, to the mobility of the phosphorous atoms and the low number of unique reflections compared to anomalous scatterers[42]. We are currently exploring ways to optimise the P-signal for P-SAD applications.

### Structural description of an intramolecular i-motif
The crystal structure formed from the ILPRC sequence 4C (8AYG) is comprised of two, independent, and inverted intramolecular i-motifs in the asymmetric unit (Fig. 3A, B). Each of these individual i-motifs is formed from four antiparallel strands held together by eight,

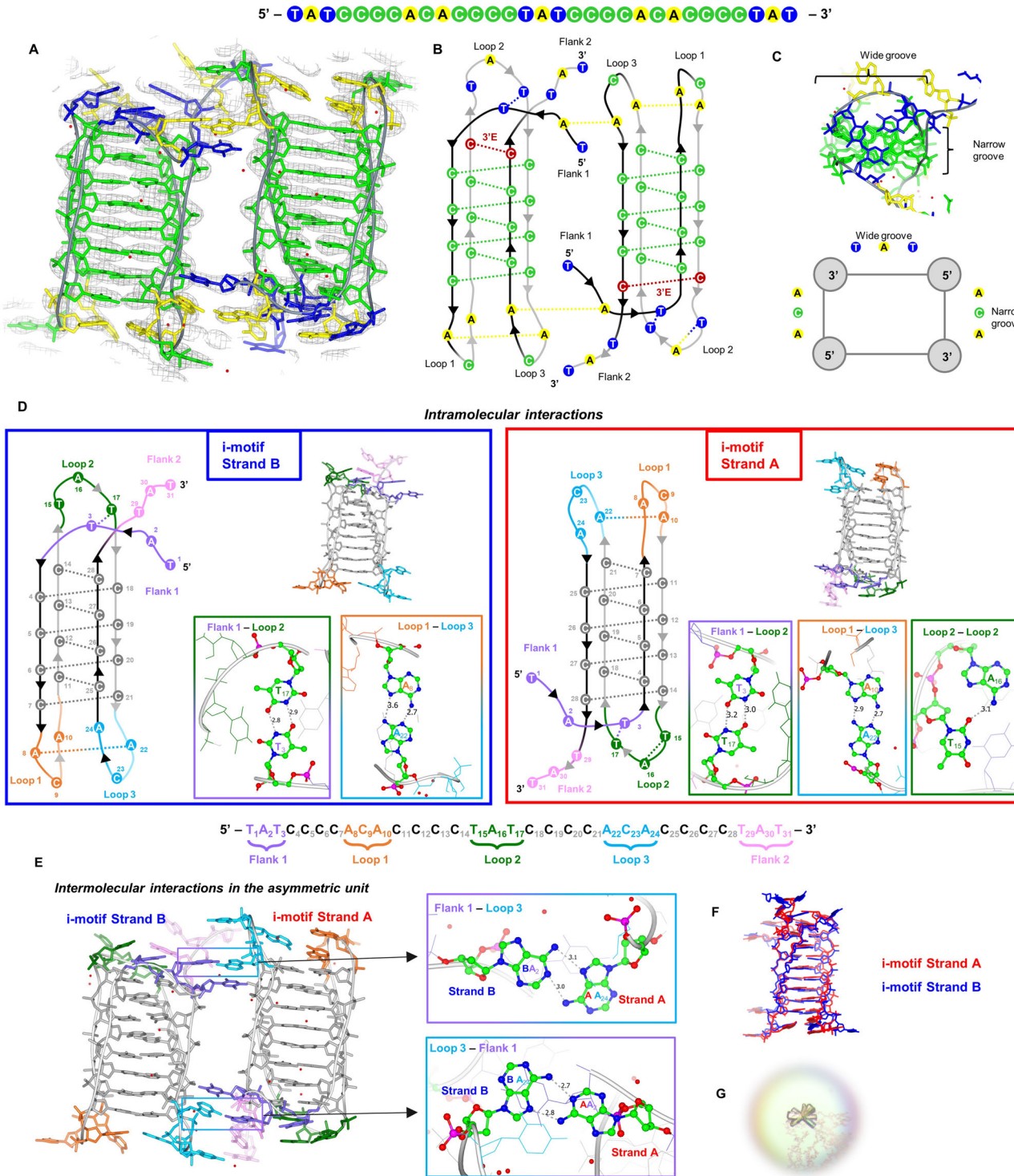

**Fig. 3 | Crystal structure, structural features and interactions of the ILPR 4C intramolecular i-motif 8AYG. A** 4C structure coloured by nucleotide type (green: C, blue: T, yellow: A, grey: backbone, red: water molecules) and *2F_obs − F_calc* electron density map contoured at 1.5 σ level (grey). **B** Schematic showing the two 4C intramolecular i-motifs as arranged in the asymmetric unit and the interactions they form. Both fold into a 3'E-topology with the outer CC pair at the 3'-end (red). **C** Top view of one 4C i-motif and schematic showing the arrangement of the TAT and ACA-loops at the wide and narrow grooves, respectively. **D** Intramolecular and

**E** Intermolecular interactions formed by the two independent intramolecular 4C i-motif strands B and A present in the asymmetric unit. Each structure and schematic is coloured based on the flank or loop position in the sequence: flank-1 (purple), loop-1 (orange), loop-2 (dark green), loop-3 (light blue), flank-2 (pink), grey (C-core). The nucleotides involved in the hydrogen-bonds shown in the boxes are coloured by atom type. All bond distances are in Å. **F** Structural comparison of the two i-motifs in the asymmetric unit by overlapping Strand-A (red) and Strand-B (blue). **G** Crystal of the 4C i-motif.

intercalated hemi-protonated cytosine-cytosine base pairs, connected by three loops (Fig. 3). The ACA-loops connect strands at the minor grooves and the middle TAT-loop at the major groove (Fig. 3C, Supplementary Table 5 and Supplementary Fig. 12). The terminal CC-base-pair is at the 3′-end, making each structure a 3′E-topology (Fig. 3B)[43].

Apart from the CC-base-pairing, other interactions within each strand include mismatched base pairs like AA and TT, which could contribute to the overall stability of the folded construct (Fig. 3D, Supplementary Table 6). In strand-A, there is an AA base-pair between loop-1 and loop-3, $A_{22}$ (loop-3) interacts with $A_{10}$ (loop-1) via two hydrogen-bonds. The topology is further stabilised in loop-2, by the $T_3$ from the flanking region, demonstrating the importance of the flanking sequence in stabilising interactions. Also, the flanking $T_3$ interaction with $T_{17}$ as a TT-base-pair stacks with the terminal CC-base-pair ($C_{14}$ and $C_{28}$). $T_{15}$ also forms a TA-pair ($T_{15}$ and $A_{16}$) via one hydrogen-bond, which then stacks on top of the TT-base-pair. While for strand-B the TT-base-pair is sandwiched between the terminal CC-base-pair and an additional TAT-triad consisting of $T_{15}$ (loop-2), $T_{29}$ (flank-2) and $A_8$ (loop-1) from the symmetry of strand-A (Supplementary Fig. 13, Table S7). Also, in strand-B the AA base-pair is formed between $A_{22}$ (loop-3) and $A_8$ rather than with $A_{10}$ (loop-1) as in strand-A.

Strand-B is similar to strand-A (Fig. 3F) with an RMSD of 2.32 Å (when flanks are excluded, nucleotides 4 to 28). A difference is that the $A_{16}$ is displaced with a symmetry-related adenine, but still stacks on top of the TT-base-pair. Also, $A_8$ displaces $A_{10}$ in the interaction with $A_{22}$, which allows $A_{10}$ to interact with a symmetry-related thymine (Fig. 3D, Supplementary Fig. 14). When only the core is included in the calculation, the RMSD is 1.04 Å, showing the high similarity between the two cores. Differences in the torsion angles and sugar puckers of the two strands, attributed to the phosphate backbone flexibility, are shown in Supplementary Table 8 and Supplementary Fig. 15.

As there are two i-motifs in the asymmetric unit, this gives a view to how more than one i-motif may interact with each other like "beads-on-a-string". There are clear interactions between flank-1 of one strand ($A_2$) and loop-3 ($A_{24}$) of the other strand (Fig. 3E). Also, there are various π-π-stacking interactions between the outer nucleotides of the TAT-flanks and an A or T in the loops which highlight the importance of flanks in crystal packing (Supplementary Table 7). Intermolecular TA- and CC-base-pairs further contribute to the crystal packing (Supplementary Figs. 14 and 16). It is important to note that the crystallisation conditions are different to those used in the solution-based experiments, with higher concentrations of DNA and other additives to initiate nucleation and crystallisation. Potentially, at lower concentrations of DNA, these intermolecular interactions might not be present, more complex higher-order conformations could occur in solution if higher concentrations are used. Nevertheless, the crystal structure has demonstrated the potential for intermolecular interactions between intramolecular i-motifs of the 4C-variant sequence. Given the ILPR is comprised of tandem repeats, these intermolecular interactions are potentially important for consideration with how ligands and nuclear proteins may interact with these structures.

No specific hydration pattern was observed at the middle of the CC core as most of the cytosine hydrogen-bond donors and acceptors are used in the formation of the CC-pairs. Some waters at the major groove were seen hydrogen-bonded with the H of N4 of the cytosine, which is not involved in the CC-base-pairing, and we observe a bridging with the phosphate O-atoms. This is in agreement with some of the other intermolecular crystal structures published e.g., 1CN0[41], 1BQJ[41], 8DHC[44], 8CXF[44], but no bifurcated hydrogen-bond to O2 of a cytosine partner was seen. Based on the use of the $F_{anom(calc)}$ maps (Supplementary Fig. 11), we can more confidently describe these peaks as water molecules and exclude sodium or chloride ions. Although limited by resolution, water molecules observed in the loops could represent potential sites for hydrogen-bond interactions, potentially useful in future ligand design or interactions with proteins.

Given the potential binding pocket revealed by $A_{16}$, which in strand-A is base-paired with $T_{15}$ and in strand-B this adenine was displaced with a symmetry-related adenine, this indicates that this site may also be interesting for potential targeting with ligands.

## Stabilising TT-base-pairs are observed in solution
Given the additional base-pairs within the crystal structure, we were interested in whether these could be observed in solution. We performed NMR spectroscopy to examine the imino-proton region, which showed a set of peaks between 15.4 and 15.8 ppm, consistent with the presence of hemi-protonated cytosines (Supplementary Fig. 17)[15]. Additional imino-proton signals at 10.9 and 11.5 ppm are consistent with the presence of TT-base-pairs[45]. Importantly, there are no signals in the region between 12.5 and 14 ppm, where the imino-proton signals from GC- and AT-base-pairing would be expected[45,46]. NMR annealing experiments (from 333 to 277 K) revealed the formation of the CC-base-pairs at 319 K, followed by the TT-base-pairs at 312 K (Supplementary Fig. 18). This indicates that the structure in solution is similar to that in the crystal structure, and the TT-base-pairs are weaker than the CC-base-pairs. A recent study looking at i-motifs using a DNA microarray containing 10,976 genomic i-motif forming sequences found that i-motifs with shorter loops ($n = 1–4$) had enhanced stability when the sequences had thymine residues directly flanking C-tracts[47]. The presence of the TT-base-pairs in both the NMR experiments and the crystal structure provides structural evidence for the reason why this is the case.

## Enhanced sampling molecular dynamics
To further explore the conformational landscape of i-motifs, we performed enhanced sampling molecular dynamics simulations. Of particular interest to us were the loops regions, which are the major contributor to the dynamics, differentiating i-motifs from each other and other nucleic acid structures; and the influence of the flanking nucleotides on the dynamics of the loops. Markov state models (MSMs) were built to study the kinetics of conformational transitions in the loop regions (Supplementary Note 1, Supplementary Figs. 19–26 and Supplementary Table 10). Upon creation of these models, both strands present a free energy landscape consisting of multiple metastable states that also explore the crystallographic conformations.

Given the interactions observed in the crystal structure originating from the flanking sequence, we looked at sequence 4C (TATCCC CACACCCCTATCCCCACACCCCTAT) and also an analogue with one base missing at the 3′-end, 4Cdel (TATCCCCACACCCCTATCCCCA CACCCCTA). Our analysis suggests that 4Cdel is far more dynamic with multiple interactions compared to 4C. Upon inspection of the structures extracted from the coarse-grained models, 4Cdel featured far more unstructured conformations in loops-1 and 3, while those in 4C seemed fairly ordered. Loop-2 in both sequences was well ordered. This would seem to suggest that slow motions in the i-motif structure are largely as a result of stabilising and for the flexibility both loop-2 and the flanking regions at the 5′- and 3′-ends. This can be visualised in the dynamics of 4C and 4Cdel. The 4C structure is longer than 4Cdel by one base (T) at the 3′-end. This extra nucleotide leads to significantly more structural ordering via π-stacking interactions within the 3′-end, which then leads to the ordered conformations observed in loop-2. Comparing this with 4Cdel, the additional stacking is not possible, and therefore, the interactions with loop-2 produces a greater number of metastable states. Since time independent components (tICs) are ordered from slowest to fastest in terms of motions, those that provide the most stability will be ordered highest than those that are faster. But still a significant number may not be fully described by the number of dimensions which the features were reduced into. This is borne out by the unstructured conformations of loops-1 and -3 in these models as opposed to the fairly ordered ones of loop-2 and the terminal regions.

The simulation data supports the hypothesis that flanking sequences are important to the stability of i-motif structure, by providing the opportunities for additional interactions that reduce conformational dynamics. This is not only important for consideration of sequence designs for in vitro experiments involving i-motifs, but also may play an important role in how small molecules and proteins can interact with i-motif structures, and their consequential effects in biology.

Here, we show that different sequence variants of the ILPR form different DNA structures in vitro and these have different effects on *in cellulo* insulin reporter expression. Importantly, not all native ILPR variants are capable of forming i-motifs and G-quadruplexes; minor changes in the sequence have been shown to give completely different structures. The crystal structure and dynamics of an intramolecular i-motif reveals that sequences within the loop regions form additional stabilising interactions. These AA-, TT- and AT-base-pairs are critical to the formation of the stable i-motif structures and reveal pockets for rational-based drug design. We also showed the importance of flanking sequence in the crystallisation of i-motif structures, through several intermolecular interactions in the crystal structure and supporting molecular dynamics. The outcomes of this work reveal the detail in the formation of stable i-motif DNA structures, with potential for rational-based drug design for compounds to target i-motifs.

## Methods

### Oligonucleotides
All tested DNA sequences were synthesised and reverse phase HPLC purified by Eurogentec (Belgium) and prepared to a final concentration of 1 mM (biophysics) or 1.5 mM (crystallography) in ultra-pure water and confirmed using a Nanodrop. Each experimental section states the DNA concentration and buffer system in which they were prepared. Biophysical samples were annealed by heating for 5 mins at 95 °C in a heating block and allowed to anneal by slow cooling to room temperature overnight. Crystallography samples were annealed using a thermocycler/PCR machine. The temperature was held at 60 °C for 5 minutes, above the melting temperature for the sequences, followed by cooling to 51 °C at a rate 1 °C/min and then to 4 °C with a rate 1 °C/3 mins. A slower cooling rate of 1 °C/3 mins was ideal to allow the sequences to fold and avoid gel formation which was observed when higher rates were used. To allow comparison, all experiments were performed in the same buffer type, though some pHs examined were outside the buffering capacity of sodium cacodylate (pH 4.0, 4.5, and 5.0). Previous control experiments have demonstrated that these samples remain at the stated pH[48].

### Circular dichroism spectroscopy
The CD spectra of the selected ILPR sequences at different pH values, were recorded on a JASCO 1500 spectropolarimeter under a constant flow of nitrogen. The C-rich ILPR samples were diluted to 10 µM in 10 mM sodium cacodylate (NaCaco, Merck) and 100 mM KCl buffer at pH values ranging from 4 to 8. G-rich ILPR samples were prepared as 10 µM in 10 mM NaCaco and either in 100 mM KCl, 100 mM NaCl, or 100 mM LiCl buffer at pH 7.0. Four spectra scans were accumulated ranging from 200 nm to 320 nm for the buffer at each pH (blank) and DNA samples and measured at 20 °C with a data pitch at 0.5 nm, scanning speed of 200 nm/min with 1 second response time, 1 nm bandwidth, and 200 mdeg sensitivity. Data was zero corrected at 320 nm and transitional pH ($pH_T$) was determined from triplicate experiments using the C-rich ILPR variants from the inflection point of the sigmoidal fit for the measured ellipticity at 288 nm and pH range.

### UV melting/annealing and thermal difference spectroscopy
UV melting/annealing and TDS experiments were performed using the Jasco V-750 UV–Vis spectrometer. The C-rich ILPR samples were annealed at 2.5 µM in 10 mM NaCaco and 100 mM KCl buffer at pH 5.5, while 2.5 µM G-rich ILPR samples were annealed in 10 mM NaCaco and 20 mM KCl buffer at pH 7.0. For the UV melting/annealing experiments, the absorbance of the samples was recorded at every 1 °C increase/decrease in three cycles at 295 nm and simultaneously at 260 nm. Initially, the samples were held at 4 °C for 10 min followed by gradual increase to 95 °C (melting). When the temperature reached 95 °C, it was held for 10 min before the process was reversed (annealing). The melting/annealing experiments were set to change the temperature at the rate of 0.5 °C/min, and absorbance was measured at 260 and 295 nm at 1 °C intervals after being held at each temperature for 5 mins. The upper and lower baseline transition was determined for each individual graph and applied on the raw data to obtain the fraction folded at 295 and 260 nm[49]. The most predominant UV melting ($T_m$) and annealing temperature ($T_a$) were identified by the first derivative method of the baseline drift corrected and fraction folded data of for each measured cycle. The average hysteresis ($T_{hysteresis}$) was determined between each cycle of melting and annealing temperatures.

The thermal difference spectra (TDS) were obtained by measuring the absorbance spectrum from 230 nm to 320 nm after 10 mins at 4 °C for the folded DNA structure and after 10 mins at 95 °C for the unfolded structure. The TDS signature is determined by subtracting the absorbance spectra of the folded structure from the unfolded structure, zero corrected at 320 nm, and then normalised to the maximum absorbance.

### Crystallography preparation of materials
The DNA sequences used in the crystallisations are provided in Supplementary Table 1. Bromination in the 4C-Br sequence was on carbon-5 of the Cytosine$_4$ and carbon-8 for the Adenine$_{16}$. As we have previously shown, substitutions at these positions do not disrupt the folded i-motif topologies[50]. Oligonucleotides were annealed in the presence of 10 mM sodium cacodylate, 18 mM NaCl at pH 5.5.

### Crystallisations
Crystallisations were achieved using the hanging-drop vapour diffusion method. For sequence 4C, the crystallisation solutions contained 49 mM sodium cacodylate buffer at pH 5.5, 45 mM NaCl, 23 % v/v 2-Methyl-2,4-pentanediol, and 2.9 mM spermine. For the drops, 1.2 µL of 0.3 mM 4C DNA solution and 1 µL crystallisation solution were combined and allowed to equilibrate. Pseudo-hexagonal crystals of the 4C sequence were formed in about two weeks at 10 °C. It was still possible to grow good quality 4C crystals at 4 °C but took longer. 4C-Br, 4Ca and 4Cb crystals were grown at 10 °C using the conditions summarised in Supplementary Table 2.

### Data collection, phasing, and structure determination
Crystals were harvested using loops, placed in oil to remove mother liquor, and then cooled in liquid nitrogen. All diffraction data were collected at the Diamond Light Source synchrotron (DLS), UK.

P-SAD data for the 4C sequence were collected at the long wavelength I23 beamline in vacuum[51]. The wavelength was tuned below the P-absorption edge ($\lambda = 5.7788$ Å, f"4) to eliminate absorption. The maximum wavelength used during this experiment was 3.9995 Å, f"2.3 to reduce absorption. Our data collections were undertaken at low-dose levels to prevent radiation damage and improve merging of multiple datasets. Good quality P-SAD data were collected but still resulted in a low P-anomalous signal as shown in Supplementary Table 3. The signal from the intrinsic P atoms alone proved insufficient for phasing and structural determination, likely due to the low number of unique reflections compared to the number of anomalous scatterers[42]. Based on data quality indicators, resolution range, completeness, $I/\sigma I$ and $R_{meas}$, the data collected at $\lambda = 2.4797$ Å, f"1.0 was selected consisting of six datasets merged to increase the resolution and thus the number of unique reflections. Data were

reduced using xia2.multiplex and subsequently re-scaled to 2.25 Å resolution with Aimless[52,53]. All data collection details are summarised in Supplementary Table 3.

Data for the 4C-Br sequence were collected at the I03 beamline. A Br-MAD experiment was conducted at three wavelengths: 0.9196 Å (peak), 0.9203 Å (inflection point) and 0.9117 Å (remote). Data was reduced and scaled as described above. Data collection details are summarised in Supplementary Table 4. The SHELX pipeline was used to determine the positions of the Br atoms and phase information[54]. These phases were then used with the merged P-SAD dataset of 4C described above. Cycles of model building and refinement were then performed using COOT[55], and REFMAC5 (CCP4i package)[56] with refinements within PHENIX (phenix.refine)[57,58] to generate a high-quality model. This initial model was used to solve the structure of the native unmodified 4C sequence and refined to 2.25 Å resolution, the final $R_{free}/R_{work}$ values are 0.2948/0.2487. Refinement statistics are included in Supplementary Table 3). Figures were generated with COOT and CCP4MG[55,59].

### Cell culture

INS-1 rat insulinoma cells (AddexBio, Catalogue number: C0018007) were cultured in RPMI-1640 medium supplemented with 10% FBS and 50 μM 2-mercaptoethanol (BME), and 1% penicillin-streptomycin which were all obtained from Gibco. The medium was changed every four days and cells were expanded when reaching 80% confluency. Experiments were carried out in cells between passages 5–9.

### Transfection of reporter gene plasmids

INS-1 cells were seeded in 6-well plates at a density of $1 \times 10^6$ cells per well in culture medium without antibiotics and transfected when 70-80% confluency was reached. The reporter plasmid pGL410_INS421 (a gift from Kevin Ferreri via Addgene; plasmid #49057; http://n2t.net/addgene:49057; RRID:Addgene_49057)[34] has a human insulin promotor regulating firefly luciferase expression, however, this promotor has only 1.5 repeats of the predominant ILPR sequence variant (1C/G). This plasmid was sub-cloned using the GenScript CloneEZ Cloning Kit (L00339) to replace the 1.5 ILPR sequence by 2.5 repeats of the variants 1C/G, 2C/G, 4C/G, or 10C/G. The PCR insert for the subcloning kit included the human insulin promotor containing either one of the four selected ILPR sequences with additional unique identifying restriction digest sites synthesised by GenScript. All plasmids including the pRL-TK renilla control reporter vector (Promega UK Ltd., E2241) were transformed in Subcloning Efficiency™ DH5α Competent Cells (Invitrogen, 18265017) using 10 ng of plasmid DNA in 50 μL of DH5α cells which were plated on Agar plates containing 100 μg/mL ampicillin. Colonies were inoculated in LB Medium (Fisher Scientific, MP Biomedicals™ 113002132) containing sterile-filtered 50 μg/mL ampicillin (Merck, 10835242001), the ampicillin concentration was reduced to increase yield of low-copy plasmids. Plasmids were extracted using QIAGEN Plasmid extraction kit following the low-copy plasmid extraction protocol. The successful ILPR variant replacement was confirmed via gel electrophoresis following restriction digests, and further confirmed by Sanger sequencing (Supplementary Note 2). The pRL-TK renilla control vector was co-transfected alongside each ILPR-controlled firefly plasmids using Lipofectamine 2000 (Thermo Fisher). The transfection of INS-1 cells was performed in sixwell plates using a total DNA amount of 3 μg with a 9:1 firefly to renilla ratio which was mixed at 1:1 DNA to Lipofectamine 2000 ratio and were incubated for 15 mins at RT before adding the transfection mix to the confluent INS-1 cells. The following day, transfected cells and untreated INS-1 cells were seeded in a white 96 well plates at a density of $1 \times 10^5$ cells per well in low glucose medium. The low medium was made up of RPMI-1640 medium (missing additives but L-glutamine and phenol red) supplemented with 2% FBS, 10 mM HEPES, 1 mM Sodium-pyruvate, 50 μM BME and 2.8 mM Glucose (all additives were obtained from Gibco). The overnight starved cells were treated with fresh low glucose

medium or high glucose medium (16.2 mM). The four different ILPR variants were treated for 4 h. The Dual Luciferase assay (Promega) was performed according to instruction manual and measured luminescence signals on SpectraMax iD3 (Molecular Devices). The resulting firefly signal was corrected with the corresponding renilla signal and final data was normalised to the firefly/renilla ratio of the low glucose signal to account for technical variation while preserve biological variation among 12 biological repeats that contained 2–3 technical repeats per experiment[60].

Data analysis and presentation were performed using GraphPad Prism version 9.0. All sets of data passed all available normal distribution tests available in GraphPad Prism and presented as mean ± SEM with indicated biological sample sizes ($n$). The statistical difference between treatments or variants was determined by one-way ANOVA and corrected with Holm–Sidak post hoc analysis.

### NMR

NMR data was recorded for the 4C ILPRC sequence. The DNA concentration was 0.7 mM in 9.1 mM sodium cacodylate buffer at pH 5.5, 91 mM KCl, and 17% $D_2O$. Samples were run at 298 K, and annealing experiments were performed between 333 and 277 K. NMR data were acquired using a 700 MHz Bruker Avance III NMR spectrometer equipped with a TCI cryoprobe operating Topspin 3.6.2. 1H 1D spectra were acquired using a perfect echo watergate experiment[61], with a 1 s recycle delay, a 35 μs delay for binomial water suppression, a 780 ms acquisition time, and a 30 ppm spectral width, and chemical shifts were referenced to the solvent. Data were processed with exponential window functions using nmrPipe[62].

### Enhanced sampling molecular dynamics

The initial ILPR i-motif crystal structure consists of two near-identical motifs packed into a dimer as a result of crystallisation. These two i-motifs were then separated into 4C (TATCCCC<u>ACA</u>CCCC<u>T</u>AT CCCC<u>ACA</u>CCCCTAT) and a modified shortened variant 4Cdel, which is missing the terminal T: (TATCCCC<u>ACA</u>CCCC<u>T</u>ATCCCC<u>ACA</u>CCCCTA). Adaptive Bandit simulations were run using the ACEMD molecular dynamics engine[63]. Full details of the protocol are listed in Supplementary Note 1. The simulation protocol was identical for both strands. The MSMs were built using the PyEMMA software[64].

### Reporting summary

Further information on research design is available in the Nature Portfolio Reporting Summary linked to this article.

## Data availability

Atomic coordinates and structure factors of the crystal structure have been deposited to the Protein Data bank under the identification code 8AYG. Files to run the MD simulations and metastable states are deposited on Zenodo: https://doi.org/10.5281/zenodo.11075102. Source data are provided with this paper.

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

## Acknowledgements

We thank Diabetes UK (awarded to Z.A.E.W. and C.J.M. to support D.G. and R.V.C., 18/0005820) and European Union's Horizon 2020 research and innovation programme (awarded to Z.A.E.W. which supported Z.D., project No 692068: BISON) for funding. We thank the DLS-CCP4 Data Collection and Structure Solution Workshop held in 2021 at the DLS, UK and the teams at beamlines I23 and I03 who helped with the data collection and structure solution. NMR was supported by the Francis Crick Institute through provision of access to the MRC Biomedical NMR Centre. The Francis Crick Institute receives its core funding from Cancer Research UK (CC1078), the UK Medical Research Council (CC1078), and the Wellcome Trust (CC1078).

## Author contributions

Z.A.E.W. and C.J.M. secured the research funding. Z.A.E.W., G.N.P., S.H. and C.J.M. conceived and designed the study; D.G., Z.D., R.V.C. and C.A.W., prepared, measured and analysed the in vitro biophysical experiments; D.G. designed and performed the reporter gene cell-based experiments; E.A. performed the crystallisations; E.A., E.O.K., and G.N.P. collected crystal data and solved the structure; D.T.S.P and S.H. performed and analysed the computational experiments. D.G., E.A., S.H. and G.N.P. prepared the figures. D.G., E.A. and Z.A.E.W. wrote the first draft of the paper; all authors contributed to the review and editing of the manuscript.

## Competing interests

The authors declare no competing interests.
