## [Peer Review File · Nature Communications]

REVIEWER COMMENTS

Reviewer #1 (Remarks to the Author):

The manuscript describes a comprehensive study of the formation of i-motif and G-quadruplex structures within the ILPR, and how these affect the regulation of insulin expression. The manuscript not only describes the thermodynamic and spectroscopic studies, but also it includes the determination of a novel three-dimensional structure of an intramolecular i-motif structure. I have focused my review on the thermodynamic and spectroscopic studies, which are closer to my research expertise.

The formation of i-motif structures by several cytosine-rich stretches within the ILPR sequence has been previously studied by Dhakal et al. (PLoS ONE, 7 (6), art. no. e39271; Biophysical Journal, 102 (11), pp. 2575 – 2584), and other authors (Organic and Biomolecular Chemistry, 3 (12), pp. 2234 – 2236). I think that these articles have not been cited in the manuscript. The authors should briefly introduce the conclusions of these previous works and discuss their own results considering them.

Line 71, “The polymorphism of ...”. The authors must explain how the eleven main variants of wild ILPR sequence listed in Table 1 were chosen. The reference 15 given here is not appropriate, as it refers to the first i-motif described back in 1993, not to ILPR variants. To my knowledge, reference 5 deals with variations in the G-rich sequence.

The pKa of cacodylic acid is around 6.3. Therefore, this is a good buffer for solutions which pH ranges from 5.0 to 7.4, approximately, according to the literature. Why using this buffer for pH values lower than 5?

Line 109. The determined Tm value for the 1C sequence is 55 oC, which is very different from the value previously reported by Dhakal et al. (37 oC) in 10 mM buffer and 100 mM KCl. The difference has been explained in the present manuscript because of the use of phosphate buffer in the previous work. However, the temperature coefficient of phosphate is quite small (-0.0028 pH units/Celsius degree, www.teknova.com). Therefore, at 95 oC, the pH of a phosphate buffered solution would be slightly smaller than at 25 oC. As example, if pH at 25 oC is 5.5, at 95 oC it would be around 5.3. I think that this fact would be translated into a slightly stabilization of the i-motif.

Looking at the melting and annealing traces depicted in Figure S1, I can see that some transitions are not fully reversible. As example, the 1C sequence shows a very different behavior at 295 nm. I wonder whether the potential baseline drift has been correctly removed from the raw data.

Table 1. How was the uncertainty of Th calculated?

Table 1. The label of Table 1 should specify pH (5.5) and T at which the corresponding structure predominates. I think that sequence 4C lacks highlighting the mutated As in bold. Also, 4G sequence in Table 2.

Table 1 and Figures S2-S4. The values of the pH-transition midpoints should be accompanied with the corresponding uncertainty values. Looking at some of the plots in Figures S2-S4, I think that

some of the pH-transition midpoints have been determined from data sets where only two points, or even one single point, lie within the transition pH range. In these circumstances, the uncertainty could be very big. On the other hand, why the 5C sequence has been fitted to two curves, whereas 11C, 10C, or even 8C, sequences have been fitted to just one single curve?

Line 151. I think that figures S6 and S9 have been swapped. This affects to the labelling and referring of figures S6-S9.

In relation to the Figure S6, there are several sequences (4G, 7G ,...) for which the measured signal at the end of the melting process is different from that measured at the beginning of the annealing process. How could this be possible? A similar comment applies to some of the plots in Figure S1 (1C, 10C, both at 260 nm).

Lines 170-178. The T_m and T_a values determined at 295 and 260 nm are very different. Could this fact be related to melting/annealing processes done too fast? Again, I wonder whether the potential baseline drift has been correctly removed from the raw data.

The crystallization conditions given in Table S2 are different from those used in thermodynamic and CD spectroscopy measurements. I suggest the addition of a short paragraph discussing the potential effects of this difference. Given that the proposed asymmetric unit shows a dimer that contains intermolecular interactions, how DNA concentration could affect to the determined T_m values?

Reviewer #2 (Remarks to the Author):

The manuscript by Guneri et al. investigated the role of DNA secondary structures in the promoter region of the insulin-linked polymorphic region (ILPR) in the regulation of Insulin expression. They postulate the formation of stable i-motif structures in the ILPR region further regulating insulin expression. However, there is no biological evidence for the formation of the i-motif structure in the ILPR region. The only physical evidence the authors have given is the luciferase construct using the different variants of ILPR but it can't support the role of i-motif formation and its further regulation of insulin expression. Indeed, the authors observed altered insulin expression using the constructs which are still weak (~1.5) and there are no proper controls in the experiment. They evaluated only two different glucose levels (2.8mM and 16.2mM) which may not be sufficient enough to come to a conclusion. The authors should validate in additional cell lines and also different levels of glucose. The authors should further prove the formation of i-motif formation in the ILPR sequence. They have done a good job solving the crystal structure of the i-motif forming sequence mentioning rational drug can be designed to alter its expression but its functional significance still remains a question inside the cell.

My main suggestion is the authors should provide more data to convince the formation of i-motif structures inside the cell and its regulation of insulin levels. Is there any interplay between i-motif and G4 in regulating the ILPR expression? The authors can consider some experimental plans to prove the same as suggested by the article by Chatterjee and colleagues.

(doi.org/10.1002/cbic.202000703) In my overall opinion, the manuscript is too weak to be considered for Nature communication, and further detailed investigation in cells is essential.

Minor comments

1. The word “Example” in Figure 1 legend is misleading. (Example circular dichroism and thermal difference spectroscopy)

Reviewer #3 (Remarks to the Author):

The manuscript by Guneri et al. investigated the role of DNA secondary structures in the promoter region of the insulin-linked polymorphic region (ILPR) in the regulation of Insulin expression. They postulate the formation of stable i-motif structures in the ILPR region further regulating insulin expression. However, there is no biological evidence for the formation of the i-motif structure in the ILPR region. The only physical evidence the authors have given is the luciferase construct using the different variants of ILPR but it can't support the role of i-motif formation and its further regulation of insulin expression. Indeed, the authors observed altered insulin expression using the constructs which are still weak (~1.5) and there are no proper controls in the experiment. They evaluated only two different glucose levels (2.8mM and 16.2mM) which may not be sufficient enough to come to a conclusion. The authors should validate in additional cell lines and also different levels of glucose. The authors should further prove the formation of i-motif formation in the ILPR sequence. They have done a good job solving the crystal structure of the i-motif forming sequence mentioning rational drug can be designed to alter its expression but its functional significance still remains a question inside the cell.

My main suggestion is the authors should provide more data to convince the formation of i-motif structures inside the cell and its regulation of insulin levels. Is there any interplay between i-motif and G4 in regulating the ILPR expression? The authors can consider some experimental plans to prove the same as suggested by the article by Chatterjee and colleagues.

(doi.org/10.1002/cbic.202000703) In my overall opinion, the manuscript is too weak to be considered for Nature communication, and further detailed investigation in cells is essential.

Minor comments

1. The word “Example” in Figure 1 legend is misleading. (Example circular dichroism and thermal difference spectroscopy)

Reviewer #4 (Remarks to the Author):

This is a very good paper indeed and deserves to be published. as is. The paper describes for the first time a crystal structure of an intramolecular DNA imotif which is a very important non-canonical DNA structure and found in the many regions of the genome where G-quadruplexes are also possible (being the complementary strand to the G4 sequences). The i-motifs are much less studied than G quadruplexes but might be an equally important part of the gene regulation system afforded by these complementary regions.

Importantly the sequences studied are those associated with insulin gene expression, which further broadens the interest of the work. The work shows that different variants of the insulin gene respond differently to glucose with the different possible structures implicated in this and this could have important implications for those with diabetes. The differential stability to mutation of the i-motif and G-quadruplex structures is also fascinating in this context.

The crystal structure of the i-motif is a very important piece of work as the key i-motif regulatory structures are intramolecular and this is the first example of that. It also reveals interesting aspects of i-motif: i-motif interactions; interactions between different nucleic acid structures are an important topic in both RNA and DNA research.

Once the type and stability of the motifs formed by both strands is established, the study of the ILPR variants using a Luciferase-based reporter gene assay is a strength of this paper. The difference for sequences 1,4 versus 2,10 in glucose response is striking. Given that some of the authors have previously contributed research on G-Quadruplex and i-Motif Structure Interdependency, showing that things that stabilise one structure tend to destabilise the other, is that also the case in this system? If so I'm intrigued if it is just the G4 or i-motif present and having an influence, or is it a combination of both forming transiently and in exchange with each other? It may not be possible to easily answer this question inside the INS-1 cells.

The work is topical and seems to have been undertaken to a high standard. The data analysis and interpretations are sound and the work described does support the conclusions and claims. There are a lot of very interesting scientific observations in this paper and it deserves to be carefully read.

Reviewer #1

1.1 The manuscript describes a comprehensive study of the formation of i-motif and G-quadruplex structures within the ILPR, and how these affect the regulation of insulin expression. The manuscript not only describes the thermodynamic and spectroscopic studies, but also it includes the determination of a novel three-dimensional structure of an intramolecular i-motif structure. I have focused my review on the thermodynamic and spectroscopic studies, which are closer to my research expertise.

We thank the reviewer for their appreciation of our interdisciplinary manuscript, which contains molecular modelling, biophysical experiments, and biological studies together to support the first crystal structure of an intramolecular i-motif.

1.2 The formation of i-motif structures by several cytosine-rich stretches within the ILPR sequence has been previously studied by Dhakal et al. (PLoS ONE, 7 (6), art. no. e39271; Biophysical Journal, 102 (11), pp. 2575 – 2584), and other authors (Organic and Biomolecular Chemistry, 3 (12), pp. 2234 – 2236). I think that these articles have not been cited in the manuscript. The authors should briefly introduce the conclusions of these previous works and discuss their own results considering them.

Thank you for the reminder of these additional papers. These were indeed in an earlier version of the manuscript, but were cut out in shortening of the introduction. We have reinstated these into this manuscript as we agree they are critical underpinning studies to the work we present:

Although the G-rich variants from the ILPR have been investigated studies on the C-rich sequences are limited to the most prevalent variant.^{19,20,23,24}

1.3 Line 71, "The polymorphism of ...". The authors must explain how the eleven main variants of wild ILPR sequence listed in Table 1 were chosen. The reference 15 given here is not appropriate, as it refers to the first i-motif described back in 1993, not to ILPR variants. To my knowledge, reference 5 deals with variations in the G-rich sequence.

Thank you for the suggestions for improvements. We have altered the text to incorporate this:

The polymorphism in the length of the regulatory promoter region of the insulin gene was suggested as a genetic marker for non-insulin-dependent diabetes in 1983.²² A follow-up population study with 298 unrelated individuals revealed that the 5' flanking region of the human insulin gene is polymorphic in both nucleotide length and sequence.¹³ Rotwein et al. reported 14 G-rich sequence variations while three of these sequences were deemed as unique and limited to 0.2% of the population. More interestingly, these variations are limited to minor changes in the loops or G-tracts from the predominant ILPR sequence.¹³ An early study, some ILPR variants were inserted as an isolated segment into a minimal prolactin promoter-luciferase construct and co-transfected with a with a known insulin related transcription factor, Pur-1, to mimic beta-cell molecular microenvironment in non-beta cells. They have shown that the overexpression of Pur-1 has different effects on the gene expression in the minimal promoter system with different ILPR variants. The highest Pur-1 affinity was associated with the most prevalent ILPR sequence and provides the initial proof of concept that the ILPR is linked to regulate expression levels of insulin.⁵ A further study focused on three of the G-rich ILPR variants and were able to correlate a relationship between the conformation of the G-quadruplex structure with binding affinity to insulin and insulin-like growth factor.²¹ Although the G-rich variants from the ILPR have been investigated studies on the C-rich sequences are limited to the most prevalent variant.^{19,20,23,24}

Here we focused on the 11 main ILPR variants of both the C-rich (ILPRC, Table 1) and the G-rich sequences (ILPRG, Table 2) to fully understand the relationship between variant sequence, structure, and function.

1.4 The pKa of cacodylic acid is around 6.3. Therefore, this is a good buffer for solutions which pH ranges from 5.0 to 7.4, approximately, according to the literature. Why using this buffer for pH values lower than 5?

We use cacodylate buffers as they are pH stable across the temperature range for melting experiments, but keep the same buffer across all biophysical experiments for consistency and the ability to compare

directly between experiments. The UV melting experiments were performed within the normal buffering range of cacodylate. For the CD measurements we appreciate that a small number of the measurements at the extreme pHs will be slightly outside the buffering range, but these are performed at room temperature and there are no further additions of other materials (which may need buffering). We have previously performed control experiments to check that the pH remains as stated outside the main buffering range, even under melting conditions, so have added some additional text to explain this in the experimental:

To allow comparison, all experiments were performed in the same buffer type, though some pHs examined were outside the buffering capacity of sodium cacodylate (pH 4.0, 4.5, and 5.0). Previous control experiments have demonstrated that these samples remain at the stated pH.⁴⁴

1.5 Line 109. The determined T_m value for the 1C sequence is 55 oC, which is very different from the value previously reported by Dhakal et al. (37 oC) in 10 mM buffer and 100 mM KCl. The difference has been explained in the present manuscript because of the use of phosphate buffer in the previous work. However, the temperature coefficient of phosphate is quite small (-0.0028 pH units/Celsius degree, www.teknova.com). Therefore, at 95 oC, the pH of a phosphate buffered solution would be slightly smaller than at 25 oC. As example, if pH at 25 oC is 5.5, at 95 oC it would be around 5.3. I think that this fact would be translated into a slightly stabilization of the i-motif.

This is also a good point and on reflection, we think the differences are likely to arise from differences in annealing, through our experience with working with these structures and previous work we have published on annealing conditions (reference 25). We have adjusted the text to reflect this:

Although the previous work was performed in phosphate buffer, which displays reduced buffering capacity at elevated temperatures,²⁰ the likely cause of the different T_m values are differences in annealing procedures.²⁵

1.6 Looking at the melting and annealing traces depicted in Figure S1, I can see that some transitions are not fully reversible. As example, the 1C sequence shows a very different behavior at 295 nm. I wonder whether the potential baseline drift has been correctly removed from the raw data.

Thank you for this important point. We have now re-analysed all (~300) the melting experiments to correct for the baseline drift. This has significantly improved the quality of the data and the updated melting and annealing temperatures are corrected in Tables 1 and 2.

1.7 Table 1. How was the uncertainty of T_h calculated?

The uncertainty of T_h is calculated as a mean +/- the standard deviation. This is stated in the table caption.

1.8 Table 1. The label of Table 1 should specify pH (5.5) and T at which the corresponding structure predominates. I think that sequence 4C lacks highlighting the mutated As in bold. Also, 4G sequence in Table 2.

Thank you for this point, this has been corrected.

1.9 Table 1 and Figures S2-S4. The values of the pH-transition midpoints should be accompanied with the corresponding uncertainty values. Looking at some of the plots in Figures S2-S4, I think that some of the pH-transition midpoints have been determined from data sets where only two points, or even one single point, lie within the transition pH range. In these circumstances, the uncertainty could be very big. On the other hand, why the 5C sequence has been fitted to two curves, whereas 11C, 10C, or even 8C, sequences have been fitted to just one single curve?

Thank you for raising this. We have repeated all the transitional pH data in triplicates and the standard deviations are now shown alongside the values in the tables.

1.10 Line 151. I think that figures S6 and S9 have been swapped. This affects to the labelling and referring of figures S6-S9.

We thank the reviewer for their eagle-eye on this and have corrected the order in the supporting information.

1.11 In relation to the Figure S6, there are several sequences (4G, 7G,...) for which the measured signal at the end of the melting process is different from that measured at the beginning of the annealing

process. How could this be possible? A similar comment applies to some of the plots in Figure S1 (1C, 10C, both at 260 nm).

This has been resolved as part of the baseline correction as per point 1.6.

1.12 Lines 170-178. The T_m and T_a values determined at 295 and 260 nm are very different. Could this fact be related to melting/annealing processes done too fast? Again, I wonder whether the potential baseline drift has been correctly removed from the raw data.

The experiments, although performed at a rate of 0.5°C/min the temperature was also held for five minutes at each temperature before recording the absorbance at 1°C intervals. To reflect this we have updated the experimental accordingly. The hysteresis from these experiments could be reduced through slower melting rates, but this would mean individual experiments would be potentially several days long at the slowest setting on our instrument. In our experience even at this significantly slower rate we still observe hysteresis for i-motif structures, so have designed our experiments to complete within a 24 hour period. We chose one melting rate for all samples, to be able to compare directly with previous studies (e.g. *Nucleic Acids Res.*, 2017, 45, 22, 13095; *Nucleic Acids Res.*, 2020, 48, 1, 55). With regards to the baseline drift- please also see response to point 1.6.

1.13 The crystallization conditions given in Table S2 are different from those used in thermodynamic and CD spectroscopy measurements. I suggest the addition of a short paragraph discussing the potential effects of this difference. Given that the proposed asymmetric unit shows a dimer that contains intermolecular interactions, how DNA concentration could affect to the determined T_m values?

It is important to note that the structure is not a dimer, we have been careful not to call it as such in the manuscript. It is a very good point about concentrations so we have added the following segment to augment the discussion:

It is important to note that the crystallisation conditions are different to those used in the solution based experiments, with higher concentrations of DNA being used and other additives to initiate nucleation and crystallisation. It is possible that at lower concentrations of DNA these intermolecular interactions might not be present, and indeed that there may be more complex higher-order conformations observed in solution if higher concentrations were used in the biophysical experiments. Nevertheless, the crystal structure has demonstrated the potential for intermolecular interactions between intramolecular i-motifs of the 4C sequence.

Reviewer #2

2.1 The manuscript by Guneri et al. investigated the role of DNA secondary structures in the promoter region of the insulin-linked polymorphic region (ILPR) in the regulation of Insulin expression. They postulate the formation of stable i-motif structures in the ILPR region further regulating insulin expression. However, there is no biological evidence for the formation of the i-motif structure in the ILPR region. The only physical evidence the authors have given is the luciferase construct using the different variants of ILPR but it can't support the role of i-motif formation and its further regulation of insulin expression.

The title of the paper represents the work and aims of the manuscript: *Structural Insights into Regulation of Insulin Expression Involving i-Motif DNA Structures in the Insulin-Linked Polymorphic Region*. The key advance is in the structural understanding and being able to link the differences in the sequences to stable formation of i-motif. The biological experiments show that similar sequences that are not capable of folding into i-motifs and G-quadruplexes are not responsive to glucose. The section is clear to state that it is both i-motif and G-quadruplexes that are required. Further work looking at i-motif and G-quadruplex variants to delineate the contribution of each structure in the ILPR is currently underway and beyond the scope of this project.

2.2 Indeed, the authors observed altered insulin expression using the constructs which are still weak (~1.5) and there are no proper controls in the experiment.

The increase in reporter activity is 2-fold for sequence 1C/G and 1.7-fold for sequence 4C/G. The increase compared to the low glucose is highly significant $p < 0.001$ in each case. This is not weak and is a clear and significant increase. Importantly, the sequences 2C/G and 10C/G do not respond at all to glucose. There are the standard controls: A *Renilla* control plasmid is used as an internal control in the luciferase-based reporter gene assay to normalize the values of the experimental reporter gene for variations that could be caused by transfection efficiency. We also used two different glucose levels to see the effects for low glucose and high glucose, which are accepted controls in the literature (see also point 2.3).

2.3 They evaluated only two different glucose levels (2.8mM and 16.2mM) which may not be sufficient enough to come to a conclusion. The authors should validate in additional cell lines and also different levels of glucose.

The use of high and low glucose conditions are completely in-line with other studies in the insulin/diabetes field, the conditions are used to mimic the change in glucose concentration in the blood following a meal. We used the exact concentrations (low glucose at 2.8 mM and high glucose at 16.2 mM) that are the standard in the field [e.g. see examples in *PLoS ONE*, 2009 4(9):e6953; *Nat. Commun.* 2022, 13, 1, 4237; *Mol Metab.* 2020, 37: 100993; *PLoS One*, 2021,16(2):e0241651. *Endocrinology.* 2021, 162(3): bqaa239; *PLoS One* 2015, 5;10(6):e0129238]. We have added some detail to explain the conditions:

The high and low glucose treatment conditions are consistent with other previous studies that measure responsiveness to glucose.³⁶⁻³⁸

As we need to measure relative expression from the insulin promoter, we need to use insulin secreting cell lines. In the field the most widely used insulin-secreting cell lines are RIN, HIT, MIN, INS-1 and TC cells. However, some of these are only poorly responsive to glucose. Culturing insulin-secreting cells is challenging and as a result the field typically also uses only one cell line in their studies. This is, indeed, in stark contrast to cancer research, which is usually the target of action of alternative DNA structures such as G-quadruplexes and i-motifs. We have used the gold-standard INS-1 cells as these have previously been used to assess levels of insulin expression in vitro but still retains normal regulation of glucose-induced insulin secretion. Finally, also given we are using reporter gene assays, we do not see how performing the experiments in other cell lines will add value.

2.4 The authors should further prove the formation of i-motif formation in the ILPR sequence. They have done a good job solving the crystal structure of the i-motif forming sequence mentioning rational drug can be designed to alter its expression but its functional significance still remains a question inside the cell.

We thank the reviewer for the recognition of the significant advance in solving the crystal structure, which is the key finding in the paper. The manuscript does not seek to prove the formation of i-motif in the ILPR sequence, which is a separate question beyond the scope of this project.

2.5 My main suggestion is the authors should provide more data to convince the formation of i-motif structures inside the cell and its regulation of insulin levels. Is there any interplay between i-motif and G4 in regulating the ILPR expression?

As stated in the abstract, here we show the first crystal structure and dynamics of an intramolecular i-motif also reveals sequences within the loop regions forming additional stabilising interactions, which are critical to the formation of the stable i-motif structures. We have also shown that the formation of i-motif and G-quadruplex structures are required for expression of insulin. These are key advances in the field in their own right. Since submitting the manuscript we have extended the study to look at other native variants and mutants of the ILPR sequence with the aim to delineate the relationship between i-motif and G-quadruplex, but this is beyond the scope of the current work described herein.

2.6 The authors can consider some experimental plans to prove the same as suggested by the article by Chatterjee and colleagues. (doi.org/10.1002/cbic.202000703)

The above article is a review of the literature – there are no similar experimental plans within the paper.

2.7 In my overall opinion, the manuscript is too weak to be considered for Nature communication, and further detailed investigation in cells is essential.

We disagree with the reviewer who considers this work weak. The manuscript is the first description of a crystal structure of intramolecular i-motif DNA, with associated dynamics and is supported by solution experiments and cell work. It is a landmark paper which will enable, finally, people interested in targeting i-motif DNA to use an accurate structure for rational based drug design of compounds to target this structure. In the manuscript the cell work supports the need and interest in targeting this structure as well. The nearest equivalent paper was recently published in Angewandte Chemie (Crystal Structure of an i-Motif from the HRAS Oncogene Promoter, *Angew. Chem. Int. Ed.* 2023;62(26):e202301666) and this was for a bimolecular structure, which is not physiologically relevant and published on its own without the dynamics, biophysics or cell work. We considered that *Nature Communications* would be an excellent place for this landmark paper to be published. It will likely be cited hundreds of times.

2.8 The word "Example" in Figure 1 legend is misleading. (Example circular dichroism and thermal difference spectroscopy)

We have changed the wording to "representative".

Reviewer #3

3.1 *This is a very good paper indeed and deserves to be published as is. The paper describes for the first time a crystal structure of an intramolecular DNA imotif which is a very important non-canonical DNA structure and found in the many regions of the genome where G-quadruplexes are also possible (being the complementary strand to the G4 sequences). The i-motifs are much less studied than G quadruplexes but might be an equally important part of the gene regulation system afforded by these complementary regions.*

We thank the reviewer for their kind words and recognition of the interdisciplinary study we have completed.

3.2 *Importantly the sequences studied are those associated with insulin gene expression, which further broadens the interest of the work. The work shows that different variants of the insulin gene respond differently to glucose with the different possible structures implicated in this and this could have important implications for those with diabetes. The differential stability to mutation of the i-motif and G-quadruplex structures is also fascinating in this context. The crystal structure of the i-motif is a very important piece of work as the key i-motif regulatory structures are intramolecular and this is the first example of that. It also reveals interesting aspects of i-motif: i-motif interactions; interactions between different nucleic acid structures are an important topic in both RNA and DNA research. Once the type and stability of the motifs formed by both strands is established, the study of the ILPR variants using a Luciferase-based reporter gene assay is a strength of this paper. The difference for sequences 1,4 versus 2,10 in glucose response is striking.*

We thank the reviewer for their overall assessment of the paper and the recognition about the difference between sequences 1/4 and 2/10. We agree this is a striking and important part of the story we have to date, which we believe will be of significant interest to the field and beyond.

3.3 *Given that some of the authors have previously contributed research on G-Quadruplex and i-Motif Structure Interdependency, showing that things that stabilise one structure tend to destabilise the other, is that also the case in this system? If so I'm intrigued if it is just the G4 or i-motif present and having an influence, or is it a combination of both forming transiently and in exchange with each other? It may not be possible to easily answer this question inside the INS-1 cells.*

We agree this is interesting. Since submitting the manuscript we have undertaken further experiments with further variants in the reporter-gene constructs to determine the relationship between the importance of the G-quadruplex and i-motif structures and the findings will be published separately.

3.4 *The work is topical and seems to have been undertaken to a high standard. The data analysis and interpretations are sound and the work described does support the conclusions and claims. There are a lot of very interesting scientific observations in this paper and it deserves to be carefully read.*

We thank this reviewer for their assessment of the manuscript. We agree that the findings are important to the field and will likely be cited hundreds of times.

REVIEWER COMMENTS

Reviewer #1 (Remarks to the Author):

In the revised version of the manuscript, the authors have answered to most of my previous questions and comments. Overall, I think that this is a very good work. There are, however, some issues that should be answered.

- In Figure S7, there are several pannels that show strange profiles for the fraction of folded DNA calculated from melting and/or annealing processes, such as those for 5G (annealing), 7G (annealing), 9G (melting and annealing), 10G (annealing), 11G (melting). This could be due to the small change of the absorbance at 295 nm along these processes. However, this should be fixed.
- This also applies to Figure S1, (panel L, sequence 11C, melting and annealing).

Reviewer #2 (Remarks to the Author):

The authors have revised their manuscript, but I am disappointed to see no additional experiments related to the cellular formation or biological functional role of DNA secondary structures in the promoter region of the insulin-linked polymorphic region (ILPR) in the regulation of Insulin expression were conducted, except for the luciferase construct from the beginning. The manuscript has undergone only minor corrections. While the luciferase construct has drawbacks that will be pointed out in the comments, it still remains a question that i-motif/G4s can control such a critical gene in controlling insulin levels? Further, multiple experiments are required before drawing a conclusion regarding the potential for rational-based drug design to alter insulin gene expression via targeting those structures without validating its formation in genome level. If the authors are unable to validate those, they need to be removed.

Abstract last time - The outcomes of this work reveal the detail information of stable i-motif DNA structures

But did the authors didn't show any evidence of stable i-motif in genome context.

As I previously stated while the authors' have solved the crystal structure of i-motif which is really interesting to the community. However, to publish in Nature communication validations and additional experiments related to the in-vivo formation of i-motif/G4s and control of insulin by these structures are necessary. A direct corelation with the Angew. Chem. Int. Ed. 2023;62(26):e202301666 cannot be justified considering the different scope of the journals. The recent trend with DNA secondary structure and its functional role in Nature communication have always presented data with their in-vivo formation by different techniques. With advance with various sequencing techniques and mapping approach now its possible to map DNA secondary

structure before strongly implying it for biological function. This work wouldn't be of great significance if the i-motif/G4 cannot fold in genomic context unless it demonstrated.

Overall G4 formation – Nature Communications volume 14, Article number: 6705 (2023), Nature Communications volume 15, Article number: 1045 (2024), Nature Communications volume 13, Article number: 6224 (2022), Nature Communications volume 14, Article number: 205 (2023).

This manuscript would be certainly interesting and potential if additional experiments related to cellular formation with essential controls are performed.

I am giving my detailed comments

1. Luciferase construct

i. The biological role of i-motif/G4s controlling the insulin gene transcription is solely based on luciferase constructs. While the data shows there is some positive role with respect to sequence, it cannot be conclusive. The plasmid-based reporter construct based on individual sequence and mutants do not directly access the transcription of ILPR in endogenous chromatin. For instance, a well-studied G4 forming sequence in c-MYC promoter where it was previously reported to block transcription using in vitro or plasmid assays was found to be positive regulator of MYC transcription in chromatin context. (<https://doi.org/10.1073/pnas.2320240121>) Hence conclusion should be drawn carefully by additional experiments in terms of genome level.

ii. Further the mechanism how it related ILPR via i-motif/G4s has not be discussed in the manuscript elaborately.

iii. The data requires a negative control i.e normal cell line lacks insulin sensitivity to state the change is mediated by insulin sensitivity mediated by the sequence alone.

2. In vivo formation of i-motif/G4s in ILPR

i. While its well known the in-vitro folding of secondary structures cannot be concluded with cellular formation especially i-motif sequence which requires a very low pH. Related to in vivo formation a recent paper mapped the genome wide formation of i-motif has been reported <https://doi.org/10.1093/nar/gkad626>. The authors can analyse/perform to first validate the formation of i-motif structures in ILPR and its positive role in transcription.

ii. Regarding the G4 there are well reported genome wide mapping techniques developed by Shankar's lab which can be used to map the G4 formation. i.e <https://www.nature.com/articles/nprot.2017.150>

Overall, the manuscript can be certainly considered if the authors could do additional experiments/analysis to address the above mentioned and further their conclusions can be strengthened.

Reviewer #3 (Remarks to the Author):

I co-reviewed this manuscript with one of the reviewers who provided the listed reports. This is part of the Nature Communications initiative to facilitate training in peer review and to provide appropriate recognition for Early Career Researchers who co-review manuscripts

Reviewer #4 (Remarks to the Author):

I am happy that the changes have addressed all the reviewers comments and in particular my own. I recommend publication as is.

Reviewer #1

- In Figure S7, there are several pannels that show strange profiles for the fraction of folded DNA calculated from melting and/or annealing processes, such as those for 5G (annealing), 7G (annealing), 9G (melting and annealing), 10G (annealing), 11G (melting). This could be due to the small change of the absorbance at 295 nm along these processes. However, this should be fixed.- This also applies to Figure S1, (panel L, sequence 11C, melting and annealing).

The reviewer is correct in that these show a very small change in absolute absorbance, which occurs when there is no melting/anneal transition in the experiment. To be explicit we have now changed the legend to denote which panels depict no melting observed and coloured these in grey as well:

e.g. for Figure S1: **Graphs D and L (in grey) highlighted in grey showed no repeatable thermodynamic profile at 295 nm.**

And for Figure S7: **Graphs C, D, J, L, and V (in grey) showed no repeatable thermodynamic profiles at 295 nm.**

Reviewer #2

The authors have revised their manuscript, but I am disappointed to see no additional experiments related to the cellular formation or biological functional role of DNA secondary structures in the promoter region of the insulin-linked polymorphic region (ILPR) in the regulation of Insulin expression were conducted, except for the luciferase construct from the beginning. The manuscript has undergone only minor corrections. While the luciferase construct has drawbacks that will be pointed out in the comments, it still remains a question that i-motif/G4s can control such a critical gene in controlling insulin levels? Further, multiple experiments are required before drawing a conclusion regarding the potential for rational-based drug design to alter insulin gene expression via targeting those structures without validating its formation in genome level. If the authors are unable to validate those, they need to be removed.

We did do additional biophysical experiments, based on reviewer 1's comments and re-analysed all the biophysical data in line with their suggestions.

We agree that more cell based studies are important and this will form part of the next stage for validation of the target. The ILPR presents significant challenges in the length and complexity of the sequence between people and we are in the process of developing appropriate models to test these. As the focus of our work is based mainly in structural biology, we have adjusted the wording to account for this in the title, which now reads:

Structural Insights into i-Motif DNA Structures in the Insulin-Linked Polymorphic Region

Also in the abstract and conclusions:

The outcomes of this work reveal the detail in formation of stable i-motif DNA structures, with potential for rational based drug design for compounds to target i-motif DNA.

Abstract last time - The outcomes of this work reveal the detail information of stable i-motif DNA structures

But did the authors didn't show any evidence of stable i-motif in genome context.

As I previously stated while the authors' have solved the crystal structure of i-motif which is really interesting to the community. However, to publish in Nature communication validations and additional experiments related to the in-vivo formation of i-motif/G4s and control of insulin by these structures are necessary. A direct corelation with the Angew. Chem. Int. Ed. 2023;62(26):e202301666 cannot be justified considering the different scope of the journals. The recent trend with DNA secondary structure and its functional role in Nature communication have always presented data with their in-vivo formation by different techniques. With advance with various sequencing techniques and mapping approach now its possible to map DNA secondary structure before strongly implying it for biological function. This work wouldn't be of great significance if the i-motif/G4 cannot fold in genomic context unless it

demonstrated.

Overall G4 formation – Nature Communications volume 14, Article number: 6705 (2023), Nature Communications volume 15, Article number: 1045 (2024), Nature Communications volume 13, Article number: 6224 (2022), Nature Communications volume 14, Article number: 205 (2023).

This manuscript would be certainly interesting and potential if additional experiments related to cellular formation with essential controls are performed.

We have carefully read the suggested papers above

- 1) Nature Communications volume 14, Article number: 205 (2023) - *Stress promotes RNA G-quadruplex folding in human cells.*

This paper is entirely different in scope, focussing on G-quadruplex (not i-motif), fully cell based and no crystallography, biophysics or modelling

- 2) Nature Communications volume 14, Article number: 6705 (2023)- *UV-induced G4 DNA structures recruit ZRF1 which prevents UV-induced senescence*

This paper is also entirely different in scope, focussing only on G-quadruplex (not i-motif), fully cell based and no crystallography, biophysics or modelling

- 3) Nature Communications volume 15, Article number: 1045 (2024)- *G-quadruplexes promote the motility in MAZ phase-separated condensates to activate CCND1 expression and contribute to hepatocarcinogenesis*

Again, this paper also focusses only on G-quadruplex (not i-motif). Again, it is fully cell-based and does use plasmid experiments, there is no crystallography or modelling.

- 4) Nature Communications volume 13, Article number: 6224 (2022) - *RNA G-quadruplex structure contributes to cold adaptation in plants*

This paper focussing only on G-quadruplex (not i-motif), is fully cell-based and does also use plasmid experiments, but again, no crystallography or modelling.

Two of the stated papers *do* still use plasmid experiments, as they are a useful model construct to assess biological activity. Importantly, all of the stated papers are about G-quadruplex, which is a much more mature field compared to i-motif. None of the above papers provide the intricate biophysical, structural and conformational detail that our paper does. The papers all reflect a different approach, in a different area, using different methods which do not include any structural detail. These are exceptional cell biology papers in their own right, but so is ours from a structural biology perspective.

I am giving my detailed comments

1. Luciferase construct

i. The biological role of i-motif/G4s controlling the insulin gene transcription is solely based on luciferase constructs. While the data shows there is some positive role with respect to sequence, it cannot be conclusive. The plasmid-based reporter construct based on individual sequence and mutants do not directly access the transcription of ILPR in endogenous chromatin. For instance, a well-studied G4 forming sequence in c-MYC promoter where it was previously reported to block transcription using in vitro or plasmid assays was found to be positive regulator of MYC transcription in chromatin context. (<https://doi.org/10.1073/pnas.2320240121>) Hence conclusion should be drawn carefully by additional experiments in terms of genome level.

We agree that the result may be different in the context of the chromatin so have added some text to account for this:

These data indicate the potential importance of the different sequence variants in the ILPR, showing the different DNA structures they form may play a role in controlling the responsiveness to glucose. Although the plasmid experiments do not directly assess transcription in endogenous chromatin, they do indicate that only small changes

in sequence can give rise to a very big difference in the structure formed and also the relative reporter expression in plasmids.

ii. Further the mechanism how it related ILPR via i-motif/G4s has not be discussed in the manuscript elaborately.

We would not like to over-analyse the data we have. We have made a clear correlation between sequence, structure and function. Further cell-based experiments to delineate the mechanism are beyond the scope of this manuscript.

iii. The data requires a negative control i.e normal cell line lacks insulin sensitivity to state the change is mediated by insulin sensitivity mediated by the sequence alone.

It is not clear what control the reviewer wants in iii and how it would make a difference to our findings.

2. In vivo formation of i-motif/G4s in ILPR

i. While its well known the in-vitro folding of secondary structures cannot be concluded with cellular formation especially i-motif sequence which requires a very low pH. Related to in vivo formation a recent paper mapped the genome wide formation of i-motif has been reported

<https://doi.org/10.1093/nar/gkad626>. The authors can analyse/perform to first validate the formation of i-motif structures in ILPR and its positive role in transcription.

ii. Regarding the G4 there are well reported genome wide mapping techniques developed by Shankar's lab which can be used to map the G4 formation. i.e <https://www.nature.com/articles/nprot.2017.150>

Overall, the manuscript can be certainly considered if the authors could do additional experiments/analysis to address the above mentioned and further their conclusions can be strengthened.

Our work is not about *in vivo* formation of i-motifs or G4s in the ILPR. It is focussed on the relationship between sequence, structure and function. We have not performed any experiments in animal models, nor have the authors in <https://doi.org/10.1093/nar/gkad626>. This is beyond the scope of our work in this manuscript and further detailed *in celluo* work will be reported separately in due course.

Reviewer #3

I co-reviewed this manuscript with one of the reviewers who provided the listed reports. This is part of the Nature Communications initiative to facilitate training in peer review and to provide appropriate recognition for Early Career Researchers who co-review manuscripts

Thank you for your comments and consideration.

Reviewer #4

I am happy that the changes have addressed all the reviewers comments and in particular my own. I recommend publication as is.

Thank you for your comments and consideration.

REVIEWERS' COMMENTS

Reviewer #1 (Remarks to the Author):

The authors have answered all my questions and remarks. In my opinion, the manuscript could be accepted as it is. Congratulations.

Reviewer #2 (Remarks to the Author):

The authors have provided revised manuscripts but again none of my comments regarding the i-motif formation inside the cell has been answered.

1. The authors answered the “Our work is not about in vivo formation of i-motifs or G4s in the ILPR” but the title stands “Structural Insights into i-Motif DNA Structures in the Insulin-Linked Polymorphic Region” and again in the abstract lines 25 – 27 states that “stable i-motif structures, with potential for rational drug design” But no evidence for i-motif structures inside the cell has been demonstrated. This is very critical because not all the i-motif forming sequences are known to fold inside the cells, which is clearly demonstrated recently - Nature Communications volume 15, Article number: 1992 (2024).

2. My previous comment is more relevant at this point, Without the evidence of i-motif formation inside the cell how can rational compound targeting those structures be helpful for controlling such an important gene? This may mislead the research community. If the authors didn't prove the i-motif formation, then they should remove such assumption.

3. The authors backup their evidence of i-motif formation with

i. Biophysical studies - which are in-vitro experiments do not necessarily validate the i-motif formation inside cell.

ii. Plasmid construct – which again doesn't validate the i-motif formation in the cell. Then how i-motif formation can be concluded?

4. The authors have again justified the previous articles I referred to as they are focusing on G4, but i-motif is also similar secondary structure. Furthermore, those articles have validated the G4 formation with high-throughput techniques either sequencing or antibodies. The plasmids constructs can only be partially justified as it never mimics the cell transcription machinery (at least in case of DNA secondary structure) which I clearly stated earlier.

5. If the authors think crystallography is more relevant, then an interesting G4 structure proposed by Butcher and colleagues Nature Structural & Molecular Biology volume 29, pages1113–1121 (2022) has clearly validated its in-vivo silencing C.elegans. The authors should at least conduct some validation in cell lines for i-motif formation or modulating them alters the gene machinery by some i-motif ligand.

6. The authors must validate the i-motif atleast in-cellulo not in-vivo (which was a mistake earlier) Then my further two suggestions for validating the i-motif/G4 formation are more relevant at this point

I- I-motif mapping - <https://doi.org/10.1093/nar/gkad626>

II- Several mapping techniques are available for G4 formation in-cellulo

Overall, it's important to show i-motif formation in-cellulo at least to strengthen their conclusion. As authors stated "A separate report in due course will be reported" cannot be justified given the importance of i-motif formation to correlate with their findings.

Reviewer #3 (Remarks to the Author):

I co-reviewed this manuscript with one of the reviewers who provided the listed reports. This is part of the Nature Communications initiative to facilitate training in peer review and to provide appropriate recognition for Early Career Researchers who co-review manuscripts

Response to Reviewers

Reviewer #1:

1.0 The authors have answered all my questions and remarks. In my opinion, the manuscript could be accepted as it is. Congratulations.

We thank this reviewer for their positive contribution to the revisions of our work.

Reviewer #2:

2.0 The authors have provided revised manuscripts but again none of my comments regarding the i-motif formation inside the cell has been answered.

We thank the reviewer for their comments on our work. We completely agree that further cell investigations on this target are warranted and important, but we disagree that further cells based experiments should be part of this manuscript, which is focussed on revealing the details in the relationship between sequence and stable structure formation. We have investigated and are still exploring the ILPR in more detail. This repetitive region of the genome is complicated and varies between people, further cell-based studies are still on-going and will be reported separately.

We also thank the reviewer for reminding us that importantly, some i-motif mapping has been performed by Sara Richter's group (<https://doi.org/10.1093/nar/gkad626>). Although this work was published after our manuscript was submitted to *Nature Communications*, it is critical we reference it within the manuscript (and is now reference 19).

2.1. The authors answered the "Our work is not about in vivo formation of i-motifs or G4s in the ILPR" but the title stands "Structural Insights into i-Motif DNA Structures in the Insulin-Linked Polymorphic Region" and again in the abstract lines 25 – 27 states that "stable i-motif structures, with potential for rational drug design" But no evidence for i-motif structures inside the cell has been demonstrated. This is very critical because not all the i-motif forming sequences are known to fold inside the cells, which is clearly demonstrated recently - Nature Communications volume 15, Article number: 1992 (2024).

Our work is not focussed on the formation of i-motifs in cells. Our paper is about the structural detail and relationship between sequence, structure and potential biological function. To be more explicit, we have changed the title to *Structural Insights into i-Motif DNA Structures in Sequences From the Insulin-Linked Polymorphic Region*. We have also made other parts of the manuscript more explicit to emphasise this:

Here we show that different sequence variants of the ILPR form different DNA secondary structures in vitro and reporter genes in cellulo indicate that insulin expression changes with formation different DNA structures.

A relationship between in vitro formation of i-motif and G-quadruplex structures within the ILPR and corresponding in cellulo reporter gene expression is determined.

Sequences that Fold into i-Motif and G-quadruplex Structures in vitro Switch Insulin Reporter Gene Transcription.

We hypothesised that the secondary DNA structures forming into i-motifs and G-quadruplexes in the ILPR are potentially binding elements to control transcription of insulin.

Given that small differences between DNA sequences resulted in different structure formation in vitro and potential function in the reporter genes in cellulo, we were interested in the potential interactions within the loops that made certain sequence variants more stable than others. In particular, we were interested in exploring the i-motif structures, where much less is known about structure compared to G-quadruplexes.

Here we show that different sequence variants of the ILPR form different DNA secondary structures in vitro and reporter gene experiments indicate these have different effects on in cellulo insulin reporter expression.

These AA, TT and AT base pairs are critical to the formation of the stable i-motif structures reveal pockets for rational based drug design.

2.2 My previous comment is more relevant at this point, Without the evidence of i-motif formation inside the cell how can rational compound targeting those structures be helpful for controlling such an important gene? This may mislead the research community. If the authors didn't prove the i-motif formation, then they should remove such assumption.

We disagree because with an intramolecular structure in hand this can be used for other non-ILPR i-motifs, in terms of structure determination (using molecular replacement) and for creation of more accurate models for other i-motifs. Understanding of how this particular i-motif is stable, will also be able to feed into other models for prediction of stable i-motifs throughout the human genome.

2.3. The authors backup their evidence of i-motif formation with i. Biophysical studies - which are in-vitro experiments do not necessarily validate the i-motif formation inside cell.

ii. Plasmid construct – which again doesn't validate the i-motif formation in the cell. Then how i-motif formation can be concluded?

We agree that i-motif formation in cells is important, but the work as it stands is also important for relating the fundamental underpinning relationship between sequence, structure and potential function.

2.4. The authors have again justified the previous articles I referred to as they are focusing on G4, but i-motif is also similar secondary structure. Furthermore, those articles have validated the G4 formation with high-throughput techniques either sequencing or antibodies. The plasmids constructs can only be partially justified as it never mimics the cell transcription machinery (at least in case of DNA secondary structure) which I clearly stated earlier.

The paper that the reviewer previously referred to is [<https://doi.org/10.1073/pnas.2320240121>] which describes how the response of G-

quadruplexes is different in native and plasmid constructs containing cMYC. This was published seven months after we submitted our manuscript to *Nature Communications*. Since this has been published we have been careful to put our work in context:

These data indicate the potential importance of the different sequence variants in the ILPR, showing the different DNA structures they form may play a role in controlling the responsiveness to glucose. Although the plasmid experiments do not directly assess transcription in endogenous chromatin, they do indicate that only small changes in sequence can give rise to a very big difference in the structure formed and also the relative reporter expression in plasmids.

Importantly however, with the example from the promoter of cMYC, although it shows that there is a difference between the endogenous and plasmid cMYC constructs, indeed both sets of experiments **do** show a change in function with formation of G-quadruplex.

2.5. If the authors think crystallography is more relevant, then an interesting G4 structure proposed by Butcher and colleagues Nature Structural & Molecular Biology volume 29, pages 1113–1121 (2022) has clearly validated its in-vivo silencing C.elegans. The authors should at least conduct some validation in cell lines for i-motif formation or modulating them alters the gene machinery by some i-motif ligand.

We thank the reviewer for bringing another paper about a G-quadruplex structure to our attention. i-Motifs and G-quadruplexes are very different in terms of their overall shape, the ways they are stabilised, the targeting of these with ligands and the maturity of the respective fields. As an example, the first intramolecular G-quadruplex crystal structure was published in Nature in 2002. Our work includes the first intramolecular crystal of i-motif and modelling, as well as biophysical and plasmid cell based studies. Our work represents a step change in our understanding of i-motif structure and deserves to be in *Nature Communications* as it stands.

2.6. The authors must validate the i-motif atleast in-cellulo not in-vivo (which was a mistake earlier) Then my further two suggestions for validating the i-motif/G4 formation are more relevant at this point

I- I-motif mapping - <https://doi.org/10.1093/nar/gkad626>

II- Several mapping techniques are available for G4 formation in-cellulo

Overall, it's important to show i-motif formation in-cellulo at least to strengthen their conclusion. As authors stated "A separate report in due course will be reported" cannot be justified given the importance of i-motif formation to correlate with their findings.

We completed this work and submitted our manuscript before Sara Richter's landmark paper was published. We have and are still exploring this region of the genome in detail, in the relevant cell types and conditions. Due to the ILPR being a type of VNTR, which varies between individuals, this work is more complex than most other non-

canonical structures in gene promoters. This work is still on-going and these studies will be reported separately at a later date.

Reviewer #3:

3.0 I co-reviewed this manuscript with one of the reviewers who provided the listed reports. This is part of the Nature Communications initiative to facilitate training in peer review and to provide appropriate recognition for Early Career Researchers who co-review manuscripts

We thank Reviewer 3 for co-reviewing the manuscript as part of their training and hope they have found the experience helpful.